



# Unraveling the mechanisms and implications of a stronger mid-Pliocene AMOC in PlioMIP2

Julia E. Weiffenbach[1], Michiel L.J. Baatsen[1], Henk A. Dijkstra[1,2], Anna S. von der Heydt[1,2], Ayako Abe-Ouchi[3], Esther C. Brady[4], Wing-Le Chan[3], Deepak Chandan[5], Mark A. Chandler[6], Camille Contoux[7], Ran Feng[8], Chuncheng Guo[9], Zixuan Han[10,11], Alan M. Haywood[12], Qiang Li[11], Xiangyu Li[13], Gerrit Lohmann[14, 15], Daniel J. Lunt[16], Kerim H. Nisancioglu[17,18], Bette L. Otto-Bliesner[4], W. Richard Peltier[5], Gilles Ramstein[7], Linda E. Sohl[6], Christian Stepanek[14], Ning Tan[7,19], Julia C. Tindall[12], Charles J. R. Williams[16,20], Qiong Zhang[11], and Zhongshi Zhang[9,13]

[1]Institute for Marine and Atmospheric research Utrecht (IMAU), Department of Physics, Utrecht University, 3584 CC Utrecht, The Netherlands
[2]Centre for Complex Systems Science, Utrecht University, 3584 CE Utrecht, the Netherlands
[3]Atmosphere and Ocean Research Institute, The University of Tokyo, Kashiwa, 277-8564, Japan
[4]National Center for Atmospheric Research, (NCAR), Boulder, CO 80305, USA
[5]Department of Physics, University of Toronto, Toronto, M5S 1A7, Canada
[6]CCSR/GISS, Columbia University, New York, NY 10025, USA
[7]Laboratoire des Sciences du Climat et de l'Environnement, LSCE/IPSL, CEA-CNRS-UVSQ Université Paris-Saclay, 91191 Gif-sur-Yvette, France
[8]Department of Geosciences, College of Liberal Arts and Sciences, University of Connecticut, Storrs, CT 06033, USA
[9]NORCE Norwegian Research Centre, Bjerknes Centre for Climate Research, 5007 Bergen, Norway
[10]College of Oceanography, Hohai University, Nanjing, China
[11]Department of Physical Geography and Bolin Centre for Climate Research, Stockholm University, Stockholm, 10691, Sweden
[12]School of Earth and Environment, University of Leeds, Woodhouse Lane, Leeds, West Yorkshire, LS2 9JT, UK
[13]Department of Atmospheric Science, School of Environmental studies, China University of Geoscience, Wuhan 430074, China
[14]Alfred-Wegener-Institut – Helmholtz-Zentrum für Polar und Meeresforschung (AWI), 27570 Bremerhaven, Germany
[15]Department of Environmental Physics and MARUM, University of Bremen, 28359 Bremen, Germany
[16]School of Geographical Sciences, University of Bristol, Bristol, BS8 1SS, UK
[17]Bjerknes Centre for Climate Research, Department of Earth Science, University of Bergen, 5007 Bergen, Norway
[18]Centre for Earth Evolution and Dynamics, University of Oslo, 0315 Oslo, Norway
[19]Key Laboratory of Cenozoic Geology and Environment, Institute of Geology and Geophysics, Chinese Academy of Sciences, Beijing 100029, China
[20]NCAS-Climate, Department of Meteorology, University of Reading, RG6 6ET Reading, UK

**Correspondence:** Julia E. Weiffenbach (j.e.weiffenbach@uu.nl)

**Abstract.** The mid-Pliocene warm period (3.264–3.025 Ma) is the most recent geological period in which the atmospheric $CO_2$ concentration was approximately equal to the concentration we measure today (ca. 400 ppm). Sea surface temperature (SST) proxies indicate above-average warming over the North Atlantic in the mid-Pliocene with respect to the pre-industrial period, which may be linked to an intensified Atlantic Meridional Overturning Circulation (AMOC). Earlier results from the

Pliocene Model Intercomparison Project Phase 2 (PlioMIP2) show that the ensemble simulates a stronger AMOC in the mid-





Pliocene than in the pre-industrial. However, no consistent relationship between the stronger mid-Pliocene AMOC and either the Atlantic northward ocean heat transport (OHT) or average North Atlantic SSTs has been found. In this study, we look further into the drivers and consequences of a stronger AMOC in mid-Pliocene compared to pre-industrial simulations in PlioMIP2. We find that all model simulations with a closed Bering Strait and Canadian Archipelago show reduced freshwater

transport from the Arctic Ocean into the North Atlantic. The resulting increase in salinity in the subpolar North Atlantic and Labrador Sea drives the stronger AMOC in the mid-Pliocene. To investigate the dynamics behind the ensemble's variable response of the total Atlantic OHT to the stronger AMOC, we separate the Atlantic OHT into two components associated with either the overturning circulation or the wind-driven gyre circulation. While the ensemble mean of the overturning component is increased significantly in magnitude in the mid-Pliocene, it is partly compensated by a reduction of the gyre component in

the northern subtropical gyre region. This indicates that the lack of relationship between the total OHT and AMOC is due to changes in OHT by the subtropical gyre. The overturning and gyre components should therefore be considered separately to gain a more complete understanding of the OHT response to a stronger mid-Pliocene AMOC. In addition, we show that the AMOC exerts a stronger influence on North Atlantic SSTs in the mid-Pliocene than in the pre-industrial, providing a possible explanation for the improved agreement of the PlioMIP2 ensemble mean SSTs with reconstructions in the North Atlantic.

# 1   Introduction

At a $CO_2$ concentration similar to today (ca. 400 ppm) (Seki et al., 2010; Pagani et al., 2010; Badger et al., 2013; Haywood et al., 2016; de la Vega et al., 2020), the mid-Pliocene warm period (mPWP, 3.264–3.025 Ma) is the most recent geological period of sustained warmth. The mid-Pliocene climate features global surface temperatures that were roughly 3°C higher than in the pre-industrial, substantially smaller ice sheets and a reduced meridional temperature gradient (Haywood et al.,

2013b, 2020). Many mid-Pliocene boundary conditions, such as the geographic position of the continents and oceans, were similar to the present day. Studying the mid-Pliocene climate can therefore provide us with knowledge that is highly relevant to understanding climate dynamics in a near future greenhouse climate (Burke et al., 2018; Tierney et al., 2019).

One component of the mid-Pliocene climate system that is of particular interest is the Atlantic Meridional Overturning Circu-

lation (AMOC). The AMOC is an essential mechanism of poleward heat transport and has a profound impact on the global climate system. It has been linked to many other components of the climate system, such as precipitation, average Northern Hemisphere temperatures, and North Atlantic storm tracks (Jackson et al., 2015). Future projections predict a decline in the AMOC as a response to global warming (Weijer et al., 2020), which would inevitably impact the European and North American climate (Jackson et al., 2015; Haarsma et al., 2015). However, proxy data suggest that the mid-Pliocene AMOC was stronger

than it is at present (Dowsett et al., 1992; Raymo et al., 1996; Ravelo and Andreasen, 2000). This seems to be supported by reconstructions of enhanced sea surface temperature (SST) warming in the North Atlantic in the mid-Pliocene (McClymont et al., 2020), presumably linked to more northward ocean heat transport by a stronger AMOC.

The Pliocene Model Intercomparison Project Phase 2 (PlioMIP2) was initiated to gain further insight into the dynamics of
the mid-Pliocene climate (Haywood et al., 2016). PlioMIP2 is an ensemble of seventeen coupled atmosphere-ocean Earth Sys-
tem Models that provide one mid-Pliocene simulation (mPWP; $Eoi^{400}$) and one pre-industrial simulation (PI; $E^{280}$) (Haywood
et al., 2016). The ensemble simulates a mid-Pliocene time slice on an interglacial peak (3.205 Ma), with an orbital forcing
similar to the present day. In addition, boundary conditions such as palaeogeography and ice sheet cover were provided based
on an updated reconstruction by the Pliocene Research, Interpretation and Synoptic Mapping (PRISM4) project (Dowsett
et al., 2016). Important differences between the pre-industrial and mid-Pliocene boundary conditions include the closure of the
Bering Strait and Canadian Archipelago as well as a strong reduction in the extent of the Greenland and West-Antarctic Ice
Sheet.

In this paper, we will look into the effect of a stronger mid-Pliocene AMOC on North Atlantic SSTs in the PlioMIP2 en-
semble. Previous analysis on average North Atlantic SSTs and the AMOC strength led to the conclusion that the reconstructed
SST warmth in the North Atlantic cannot be attributed to an intensified mid-Pliocene AMOC (Zhang et al., 2021b). However,
it has been shown the response of North Atlantic SSTs to the AMOC varies among coupled general circulation models (Ba
et al., 2014; Kim et al., 2018) as there are many different variables that affect SSTs (Zhang et al., 2019), such as aerosols
and clouds (Booth et al., 2012; Feng et al., 2019) or atmospheric stochastic forcing (Clement et al., 2015). Here, we will
examine reconstructed mid-Pliocene SSTs at six sites in the North Atlantic and compare these to PlioMIP2 ensemble results.
The reconstructed SSTs originate from the marine isotope KM5c timeslice (3.204-3.207 Ma), which is a warm interval in the
mid-Pliocene during which orbital forcing was similar to present day (Haywood et al., 2013a). Furthermore, we will investi-
gate whether there is a difference in the degree to which the AMOC influences North Atlantic SSTs in the mid-Pliocene and
pre-industrial.

In an analysis of the AMOC in the PlioMIP2 ensemble, Zhang et al. (2021b) reported a stronger AMOC in all mid-Pliocene
simulations compared to the pre-industrial. It is likely that the strengthening originates from the closure of the Bering Strait
and Canadian Archipelago in the mid-Pliocene. Earlier work shows that closing these Arctic gateways leads to a strengthened
AMOC due to altered freshwater fluxes from the Arctic into the North Atlantic (Otto-Bliesner et al., 2017). Several PlioMIP2
model groups have identified the changes in paleogeography, specifically the Arctic gateways closure, to be the cause of a
stronger simulated mid-Pliocene AMOC (Hunter et al., 2019; Chan and Abe-Ouchi, 2020; Tan et al., 2020; Feng et al., 2020;
Stepanek et al., 2020; Baatsen et al., 2022). In this study, we will consider whether the closure of the Arctic gateways in the
mid-Pliocene is the main driver behind the stronger mid-Pliocene AMOC in the PlioMIP2 ensemble. If this the case, geo-
graphic boundary conditions may present a significant forcing on AMOC strength during the mid-Pliocene, which should then
be considered a non-analog feature for a future warm climate.

All PlioMIP2 models simulate an intensified AMOC - but the response of the Atlantic northward ocean heat transport (OHT)
to this strengthening has been shown to be inconsistent (Zhang et al., 2021b). We will look further into the OHT response





by partitioning the Atlantic OHT into a component associated with the overturning circulation and a component associated
with the wind-driven gyres. Classical scaling shows that the OHT of an overturning cell is linearly proportional to the strength
of the overturning and the temperature difference between the northward and southward flowing water in the cell (Vallis and
Farneti, 2009). However, the total OHT is a sum of the heat transport associated with the overturning circulation as well as
the heat transport associated with the wind-driven gyre circulation. This may explain why no one-to-one relationship has been
found between the total OHT and AMOC strength in the PlioMIP2 ensemble (Zhang et al., 2021b). The wind-driven gyre
OHT is closely coupled to the atmospheric heat transport (AHT) (Vallis and Farneti, 2009). This results in a complex interplay
between the OHT and AHT (Rose and Ferreira, 2012; Yang and Dai, 2015), and therefore also between the two components
of the OHT itself, where a high degree of compensation between the two is found especially in the North Atlantic (Farneti and
Vallis, 2013). It has previously been shown that this partitioning can shed light on which oceanic processes are dominating the
OHT, thereby also identifying possible consequences for the climate system (Ferrari and Ferreira, 2011; Yang et al., 2015)


In Section 2, we will first introduce the PlioMIP2 models and the methods used to analyze the mid-Pliocene AMOC. We show
results of our analysis of the mid-Pliocene North Atlantic SSTs in Section 3.1, followed by an illustration of how changes in
freshwater transport and salinity drive a stronger mid-Pliocene AMOC compared to the pre-industrial in Section 3.2. Next, we
will study the consequences of a stronger mid-Pliocene AMOC, specifically how this affects OHT by the overturning circula-
tion (Section 3.3) and transports by the subtropical gyre (Section 3.4). We will provide a discussion of our results in Section 4
and present our conclusions in Section 5.

## 2 Data and methods

### 2.1 PlioMIP2 models

For this study, complete datasets for analysis are available from 15 out of 17 models participating in PlioMIP2. The models
are listed in Table 1 along with their institute and reference to the individual model description. All models have performed a
pre-industrial ($E^{280}$) simulation at a $CO_2$ concentration of 280 ppm (models participating in CMIP6 at 284.3 ppm, see Table
1) and a mid-Pliocene ($Eoi^{400}$) simulation at a $CO_2$ concentration of 400 ppm and with boundary conditions implemented as
described in Haywood et al. (2016). The exception is HadGEM3, where the pre-industrial land-sea mask and bathymetry is
also used in the mid-Pliocene simulation (Williams et al., 2021). For this reason, the model is excluded from any multi-model
mean (MMM) and ensemble standard deviation calculations. It is included in figures with individual model results and shown
as an unfilled marker in scatter plots. It serves as an additional reference of a model with higher than pre-industrial $CO_2$ levels
but limited geographic changes.





**Table 1.** Overview of PlioMIP2 models used in this study.

| Model name | Institute | Reference |
|---|---|---|
| CCSM4 | NCAR, USA | Feng et al. (2020) |
| CCSM4-UoT | University of Toronto, Canada | Chandan and Peltier (2017) |
| CCSM4-Utr | IMAU, Utrecht University, The Netherlands | Baatsen et al. (2022) |
| CESM1.2 | NCAR, USA | Feng et al. (2020) |
| CESM2[1] | NCAR, USA | Feng et al. (2020) |
| COSMOS | AWI, Germany | Stepanek et al. (2020) |
| EC-Earth3-LR[1] | Stockholm University, Sweden | Zhang et al. (2021a) |
| GISS2.1G[1] | GISS, USA | - |
| HadCM3 | University of Leeds, UK | Hunter et al. (2019) |
| HadGEM3[1,2] | University of Bristol, UK | Williams et al. (2021) |
| IPSL-CM5A | LSCE, France | Tan et al. (2020) |
| IPSL-CM5A2 | LSCE, France | Tan et al. (2020) |
| IPSL-CM6A[1] | LSCE, France | Lurton et al. (2020) |
| MIROC4m | JAMSTEC, Japan | Chan and Abe-Ouchi (2020) |
| NorESM1-F[1] | BCCR, Norway | Li et al. (2020) |

[1] CMIP6 model with pre-industrial 1850 $CO_2$ at 284.3 ppm.

[2] Pre-industrial land-sea mask in both simulations.

## 2.2 Proxy data

To compare mid-Pliocene model results with reconstructed data, we use SST proxy data from a 30k year interval on the KM5c
time slice by Foley and Dowsett (2019) and McClymont et al. (2020). The PRISM4 dataset (Foley and Dowsett, 2019) is a
collection of $U_{37}^{k'}$ proxies (calibrated using the Müller et al. (1998) method). The dataset by McClymont et al. (2020) includes
both $U_{37}^{k'}$ proxies (calibrated using the BAYSPLINE method) as well as foraminifera Mg/Ca proxy SST reconstructions. In
this study, SST reconstructions from six different sites between 30°N and 70°N in the North Atlantic have been used. For
comparison to model anomalies, the NOAA ERSST5 dataset (Huang et al., 2017) for the years 1870-1899 is used as the
observational pre-industrial SST dataset.





## 2.3 Data analysis

All models except IPSL-CM5A and IPSL-CM5A2 have provided 100 years of annual model data for both $E^{280}$ and $Eoi^{400}$. IPSL-CM5A and IPSL-CM5A2 have provided 50-year averages only, so they are excluded from analysis that requires annual data.


Percentage differences between the mid-Pliocene and pre-industrial are computed relative to the pre-industrial. An anomaly is defined as the difference between the mid-Pliocene and pre-industrial (mPWP-PI; $Eoi^{400}$-$E^{280}$), unless stated otherwise. Standard deviations of individual model means are calculated using annual data. Standard deviations of MMM anomalies are calculated using the individual model $Eoi^{400}$-$E^{280}$ anomalies.


Data provided by individual modelling groups includes the ocean potential temperature, ocean meridional velocity, ocean salinity, the Atlantic meridional streamfunction, the total Atlantic OHT and the atmospheric zonal and meridional wind at 1000 hPa. In addition, the atmospheric surface freshwater flux (precipitation minus evaporation; PmE) fields for all models provided by Han et al. (2021) have been used. SST, SSS, PmE and atmospheric wind fields have been interpolated to a $1°\times1°$

regular grid using bilinear interpolation. All calculations for Atlantic ocean heat transport and freshwater transport have been performed on model native grids, except for COSMOS where it is done on the regular interpolated grid. These computations have been done using Atlantic region masks provided by the individual modelling groups. The mean AMOC strength is calculated from the 100-yr mean Atlantic meridional overturning streamfunction and the yearly AMOC strength from the annual mean Atlantic meridional overturning streamfunction. The AMOC strength is defined as the maximum value of the

Atlantic meridional overturning streamfunction north of the equator and below 500 m depth. Potential density is calculated from potential temperature and salinity using the TEOS-10 equation of state (Roquet et al., 2015).

### 2.3.1 Ocean heat transport

The OHT is defined at every latitude $y$ as:

$$OHT = \rho_0 c_p \int \int_{x_E}^{x_W} vT \, dx \, dz \tag{1}$$

Here, $v$ is the meridional ocean velocity and $T$ the ocean potential temperature of each grid cell of $dx$ in longitude and $dz$ in depth. The y-dependency of the OHT is omitted in the notation. For the Atlantic OHT, integration is performed zonally between the eastern and western bounds of the Atlantic Ocean, respectively $x_E$ and $x_W$. The constants $\rho_0$ and $c_p$ are the average density and specific heat of sea water, respectively. These are set at $\rho_0 = 1.026 \cdot 10^{-3}$ kg cm$^{-3}$ and $c_p = 3996$ J K kg$^{-1}$. In this study, the total OHT is calculated from the 100-year mean 3D meridional heat transport ($vT$) field. This total OHT can be partitioned

into a mean flow $OHT^M$ and a transient component $OHT^T$ (Viebahn et al., 2016):

$$OHT = OHT^M + OHT^T \tag{2}$$





where the time mean component $OHT^M$ and transient component $OHT^T$ are

$$OHT^M = \rho_0 c_p \int \int_{x_E}^{x_W} \overline{v}\,\overline{T}\, dx\, dz \qquad (3)$$

$$OHT^T = \rho_0 c_p \int \int_{x_E}^{x_W} \overline{v'T'}\, dx\, dz \qquad (4)$$

Here, $v = \overline{v} + v'$ and $T = \overline{T} + T'$ where $\overline{v}$ and $\overline{T}$ are the 100-year mean and $v'$ and $T'$ the transient components of the respective $v$ and $T$ fields. As the 100-year mean $vT$, $\overline{v}$ and $\overline{T}$ are available as model output, this allows us to deduce $\overline{v'T'}$.

Next, we are able to separate $OHT^M$ from (3) into two separate components associated with either the zonal mean flow or the azonal flow. This will approximately separate $OHT^M$ in heat transport that can be attributed to the overturning cir-
culation (i.e. the zonal mean flow) and heat transport that results from wind-driven gyre circulation (i.e. the azonal flow).

$$OHT^M = OHT_{ov} + OHT_{az} \qquad (5)$$

Following Dijkstra (2007) and Viebahn et al. (2016):

$$OHT_{ov} = \rho_0 c_p \int \int_{x_E}^{x_W} \langle\overline{v}\rangle\langle\overline{T}\rangle\, dx\, dz \qquad (6)$$

$$OHT_{az} = \rho_0 c_p \int \int_{x_E}^{x_W} \langle\overline{v}^*\overline{T}^*\rangle\, dx\, dz \qquad (7)$$

Here, $\langle\overline{v}\rangle$ and $\langle\overline{T}\rangle$ are the zonal average of $\overline{v}$ and $\overline{T}$, and $\overline{v}^*$ and $\overline{T}^*$ are the azonal components such that $\overline{v} = \langle\overline{v}\rangle + \overline{v}^*$ and $\overline{T} = \langle\overline{T}\rangle + \overline{T}^*$.

This separation into overturning and gyre components can only be done for the time mean component as instantaneous ve-
locity fields are not available. Therefore, the total OHT will always contain a transient component $OHT^T$ in addition to the mean overturning $OHT_{ov}$ and gyre components $OHT_{az}$:

$$OHT = OHT_{ov} + OHT_{az} + OHT^T \qquad (8)$$

The variability contained in the transient component $OHT^T$ is primarily a result of the seasonal cycle and is therefore largely unaffected by the choice of averaging period (Viebahn et al., 2016). Only in areas of high eddy activity, primarily the Southern
Ocean, the transient component is significant (Yang et al., 2015). Supplementary figure S5 shows a comparison between the total Atlantic OHT ($OHT$) and the time mean Atlantic OHT ($OHT^M = OHT_{ov} + OHT_{az}$), where no significant discrepancy is found for most models. The exceptions are the COSMOS, HadCM3, GISS2.1G, MIROC4m and NorESM1-F. In COSMOS





and HadCM3, this difference most likely occurs as a result of using surface heat fluxes at the sea surface to calculate the total OHT. As this approach requires assuming that there is no change in ocean heat storage and that all heat that vertically enters the

ocean through the surface will be transported horizontally by the circulation (Yang et al., 2015), this may indicate that either or both of these conditions are not fulfilled in these models. For COSMOS, the absence of thermal equilibrium has been explicitly shown by Stepanek et al. (2020) and Lohmann et al. (2022). Therefore, we have used $OHT^M$ that is calculated from the individual velocity and temperature fields as the total OHT for COSMOS and HadCM3. In addition, the calculated overturning OHT component from MIROC4m and COSMOS is quite noisy due to interpolation in the original velocity, temperature or

region fields. This output is smoothed using a 5º running mean.

For models with a curvilinear grid, the transport calculations on a native grid lead to a degree of error at higher latitudes due to the disalignment of the meridional velocity with the y-direction of the grid cells. For this reason, the OHT components are only calculated up to 65°N for all models.

### 2.3.2  Freshwater transport

The freshwater transport $F$ is calculated for all models simulations using the 100-year mean meridional ocean velocity $\overline{v}$ and salinity $\overline{S}$ fields:

$$F = -\frac{1}{S_0} \int \int \overline{v}(\overline{S} - S_0)\, dx\, dz \tag{9}$$

where $S_0$ is the reference salinity, which is set to be the average Atlantic Ocean salinity for every individual simulation. The

integration is performed over the ocean depth and Atlantic basin width as for the Atlantic OHT and calculated up to 65°N due to curving of curvilinear grids. The freshwater transport is also calculated for two sections of the Atlantic Ocean at 62°N west and east of Greenland, respectively referred to the Labrador Sea and Greenland Scotland Ridge (GSR) freshwater transports. Freshwater transport through the Bering Strait in the pre-industrial is also computed.

In order to analyze the separate effect of the overturning and wind-driven gyre circulation on the Atlantic freshwater transport, the same separation of components is done as for the OHT, following Dijkstra (2007):

$$F_{ov} = -\frac{1}{S_0} \int \int \langle \overline{v} \rangle (\langle \overline{S} \rangle - S_0)\, dx\, dz \tag{10}$$

$$F_{az} = -\frac{1}{S_0} \int \int \langle \overline{v}^* \overline{S}^* \rangle\, dx\, dz \tag{11}$$

The total freshwater transport does not include the transient term in our study due to the use of 100-year mean velocity and

salinity fields. This transient term is caused by seasonal variations in gyre circulation, similar to the transient OHT component, and surface salinity as well as baroclinic mesoscale eddies at the boundary of the subtropical and subpolar gyres (Treguier et al., 2012). We neglect the transient term, as it was shown to be small at a model resolution of approximately 1° (Jüling et al., 2021).





## 3 Results

Figure 1 gives an overview of the AMOC strength and its variability in the individual mid-Pliocene and pre-industrial simulations. We find that all models except CESM1.2 and HadGEM3 simulate a stronger AMOC in the Pliocene. This is a slightly different result from Zhang et al. (2021b), where CESM1.2 did simulate a stronger AMOC in the Pliocene. Discrepancies between AMOC strength reported by Zhang et al. (2021b) and Figure 1 can be attributed to a difference in the 100-year time interval of the model data.


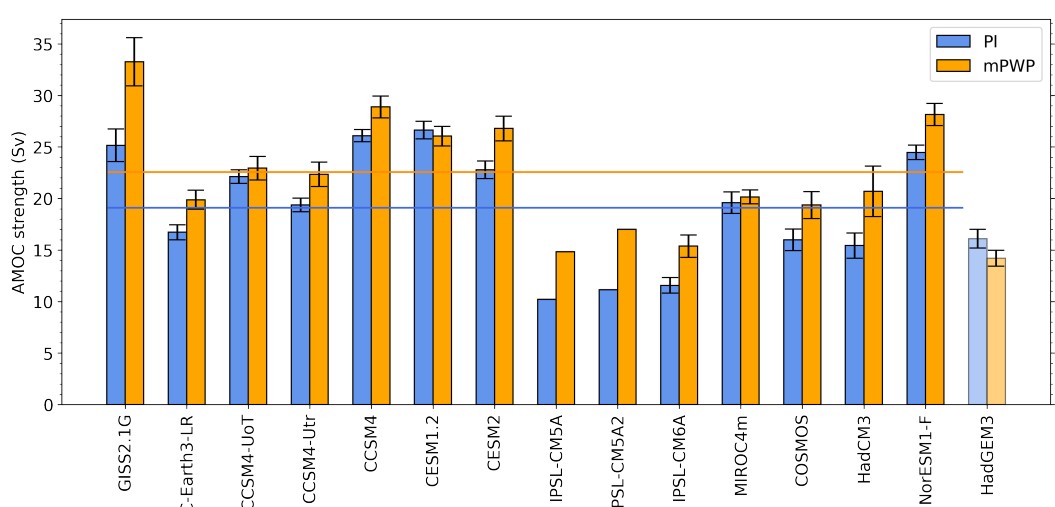

**Figure 1.** Individual model AMOC strength in the pre-industrial (blue) and mid-Pliocene (orange). Error bars indicate one standard deviation from the time-mean AMOC strength computed using 100 years of yearly data. The horizontal lines indicate the multi-model mean AMOC strength.

The error bars in Figure 1 show the standard deviation of the annual AMOC strength from the 100-year mean. The error bars for CESM1.2 show that the decrease in AMOC strength is not significant. This is also the case for the increase in AMOC strength in CCSM4-UoT and MIROC4m. For all other models included in this study, the intensified AMOC in the mid-Pliocene compared to the pre-industrial is significant. The only model simulating a significant decrease in AMOC strength is HadGEM3, 210 presumably related to the pre-industrial land-sea mask configuration in the mid-Pliocene simulation as discussed in Zhang et al. (2021b). Also note that for the majority of the models, the variability in the AMOC strength increases in the mid-Pliocene, with the exception of MIROC4m and HadGEM3.





## 3.1 North Atlantic SSTs

### 3.1.1 Comparison of models with proxy data

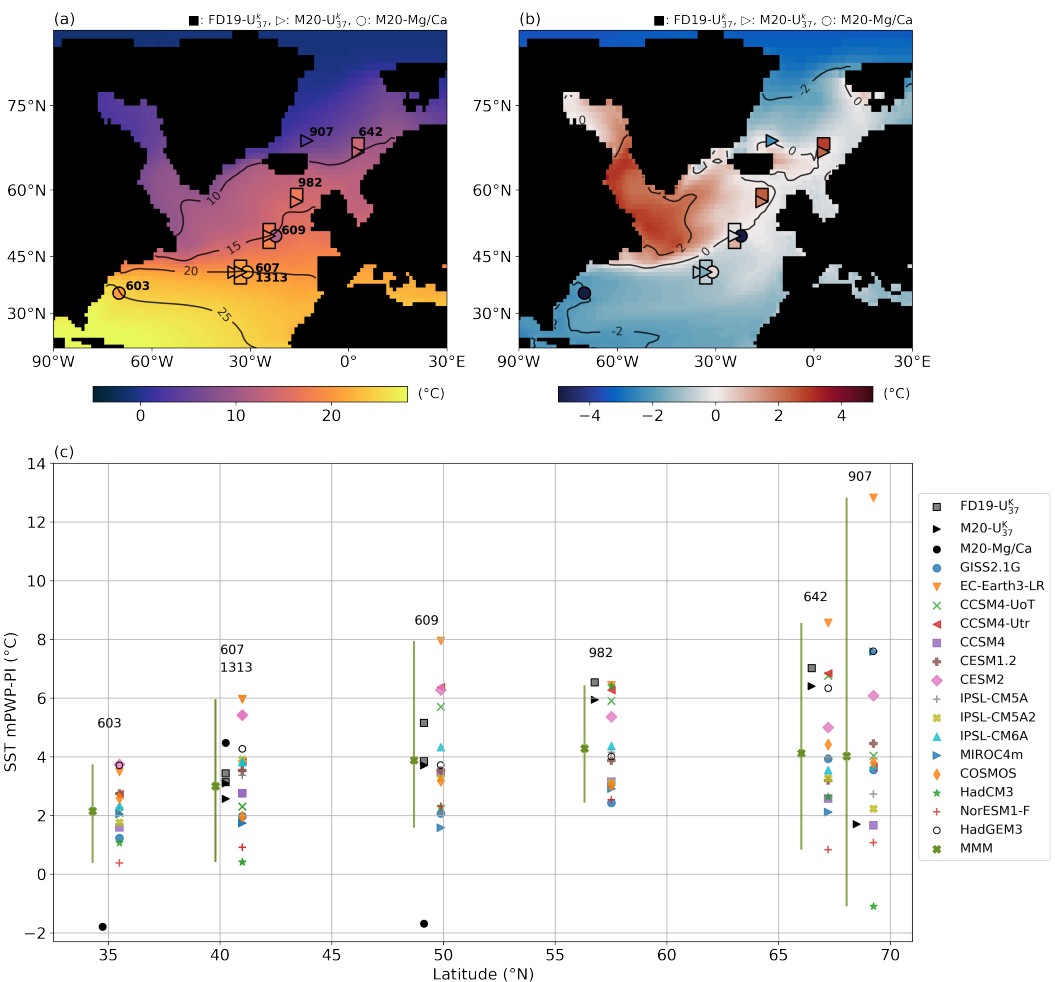

**Figure 2.** (a) Mid-Pliocene MMM and reconstructed North Atlantic SST. (b) MMM and reconstructed North Atlantic SST anomaly (mPWP-PI) with respect to the average MMM North Atlantic SST (30°N-70°N). (c) Individual model SSTs and reconstructed SSTs at six proxy locations in the North Atlantic. The vertical line shows the model spread and the MMM (excluding HadGEM3) is indicated by a cross. FD19-$U_{37}^{k'}$ refers to the Foley and Dowsett (2019) data, M20-$U_{37}^{k'}$ and M20-Mg/Ca to the McClymont et al. (2020) data. SST proxy anomalies are computed with pre-industrial ERSST5 data (Huang et al., 2017). The code to read and plot the proxy SST data has been adapted from Oldeman (2021)

Figure 2 compares the MMM and individual models' SSTs to $U_{37}^{k'}$ SST proxy data from Foley and Dowsett (2019) and $U_{37}^{k'}$ and Mg/Ca proxy data from McClymont et al. (2020) at six different North Atlantic sites. These sites are chosen based on their lo-





cation between 30-70°N, excluding sites in proximity to the Mediterranean Sea. Figure 2a shows the SSTs in the mid-Pliocene and Figure 2b the SST mPWP-PI anomalies with respect to the MMM 30°N-70°N average North Atlantic mPWP-PI SST

anomaly. Red colored regions in Figure 2b indicates above average warming and blue colored regions below average warming with respect to the average 30°N-70°N North Atlantic warming. This allows for a closer inspection of regional differences in extratropical North Atlantic warming. The modelled SST values at the different proxy sites can be found in Supplementary Table S1. Figure 2b indicates that the strongest amplified North Atlantic warming in the MMM mid-Pliocene SSTs occurs north of 45°N and west of 30°W, an area where there is currently no proxy data available for the KM5c timeslice. Most proxy

data originate from locations that are in the white area in Figure 2b, meaning that the warming simulated there is approximately equal to the extratropical North Atlantic mean warming.

The mPWP-PI anomalies of the reconstructed SSTs and MMM SSTs are plotted in Figure 2c, along with the individual model SST anomalies. We find that the discrepancy between reconstructed and MMM SST anomalies is smallest for sites 607/1313

and 609, with the exception of the relatively cold Mg/Ca reconstruction at site 609. The sites where reconstructions indicate the strongest mid-Pliocene warming are sites 982 and 642. These sites are also the only two sites that fall within the area where MMM warming is higher than the North Atlantic 30-70°N average, as seen in Figure 2b, although they are both at the edge of this area. The sites north of 60°N show a greater discrepancy between reconstructions and the MMM, as well as a relatively large model spread. The reconstructed SST at site 603 also does not agree with the MMM and falls outside the range of model

SSTs, which may be related to its location in the highly variable Gulf Stream region. This region including the Gulf Stream separation is generally not well resolved in low resolution general circulation models (Bryan et al., 2007; Saba et al., 2016; Schoonover et al., 2017).

### 3.1.2 Effect of the stronger mid-Pliocene AMOC

Even though we observe above average warming of SSTs in the northwestern North Atlantic (Figure 2b), it has been shown

that average North Atlantic temperatures do not respond consistently to a stronger mid-Pliocene AMOC (Zhang et al., 2021b). To further examine the effect of a stronger mid-Pliocene AMOC on North Atlantic SSTs, we show correlation maps between annual mean AMOC strength and SSTs at every grid point for both the pre-industrial and mid-Pliocene in Figure 3. Grid points that are colored red show positive correlation between the SST and the AMOC strength, with positive correlations greater than 0.4 indicated by a white contour line.


The majority of the PlioMIP2 models show that the effect of the AMOC on North Atlantic SSTs is different in pattern in the mid-Pliocene than in the pre-industrial, where for most models there is a relatively larger area of significant positive correlation in the mid-Pliocene (see Supplementary Table S2). Furthermore, the positive correlation with SSTs is higher in the mid-Pliocene, with the exception of CCSM4, CESM1.2, CESM2, NorESM1-F, IPSL-CM6A and HadGEM3. CCSM4 and

CESM2 do not have a stronger correlation between North Atlantic SSTs and the AMOC strength, but their maps do show a shift in the spatial pattern of correlation. The same is true for HadGEM3, which does not employ a mid-Pliocene land-sea



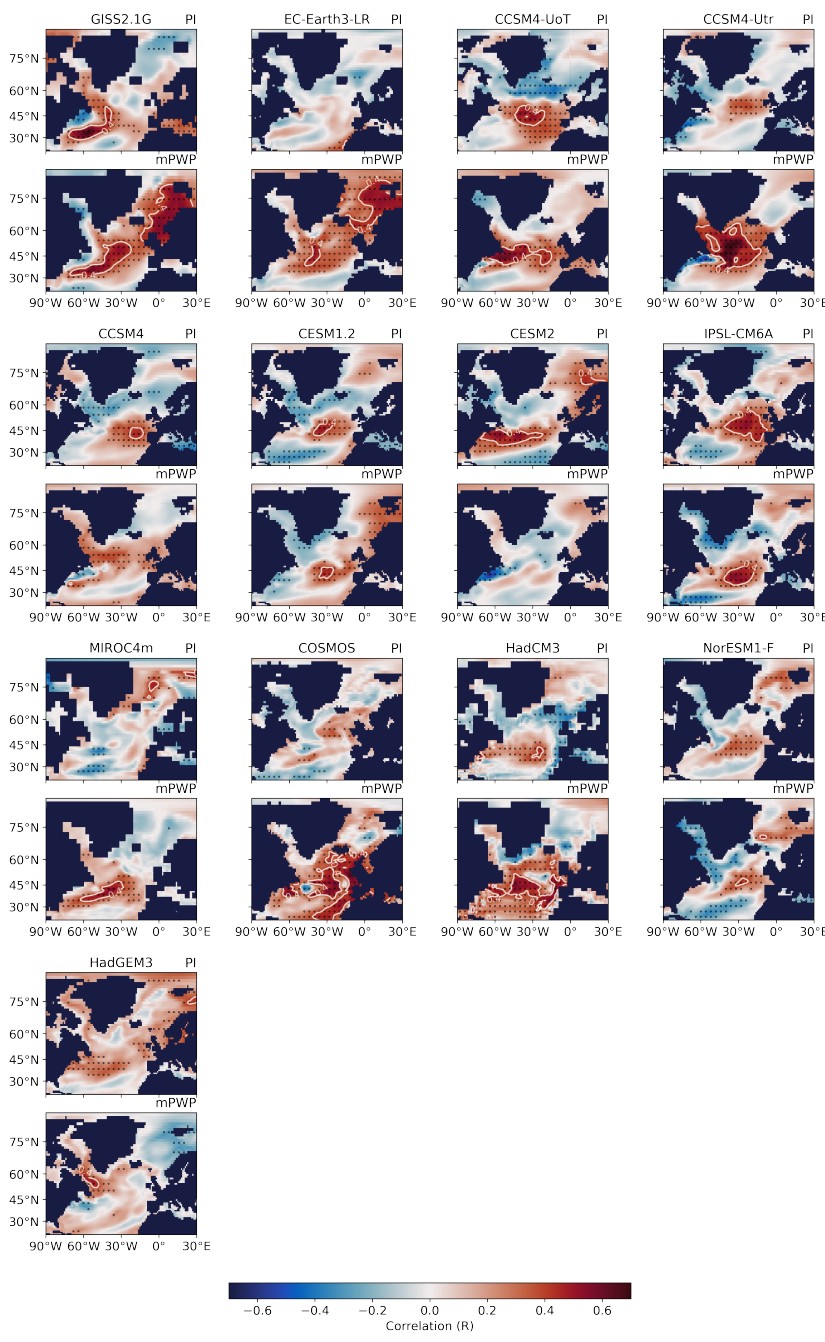

**Figure 3.** Correlation between the annual AMOC strength and annual SSTs for individual models. The top panel for each model shows the results for the pre-industrial and the bottom panel for the mid-Pliocene. Both the AMOC strength and SSTs have been linearly detrended before correlating. Stippling indicates significance at the 5% confidence level. White contours show a positive correlation of $R = 0.4$.





mask and simulates a weaker mid-Pliocene AMOC. In CESM1.2, which has a weaker AMOC in the mid-Pliocene, as well as NorESM1-F and IPSL-CM6A, the correlation remains similar in strength and pattern. The results suggest that for the majority of the models, the stronger mid-Pliocene AMOC exerts a stronger influence on North Atlantic SSTs than in the pre-industrial.
This would imply that a stronger mid-Pliocene AMOC also corresponds to higher North Atlantic temperatures, although the extent of this relation and the specific area where this is the case could be highly model dependent.

## 3.2 Density-driven increase in AMOC strength

The AMOC is known to be affected by meridional gradients in density (Rahmstorf, 1996; Thorpe et al., 2001; Stouffer et al., 2006; Wang et al., 2010) where the relatively high density in the North Atlantic is considered to be a driver of deep convection
that constitutes the sinking branch of the AMOC in the high northern latitudes. As the AMOC becomes stronger in almost all mid-Pliocene simulations relative to the pre-industrial, we expect this strengthening to scale with an increase in the meridional density gradient. Figure 4a shows the zonal mean MMM potential density of the top 1 km of the Atlantic Ocean, where it is clear that most of the Atlantic becomes less dense. This is also indicated by Figure 4b, where the potential density from Figure 4a is averaged over the top 1 km. However, for the higher northern latitudes, approximately 40-80°N, we observe that
the potential density decreases by about 0.1 kg m$^{-3}$, which is substantially less than in the rest of the Atlantic where the decrease is approximately 0.3 kg m$^{-3}$. This results in an overall increase in the meridional density gradient, which is defined as the difference between the 50-70°N average and 10-30°S average. We have plotted the meridional density gradient anomaly against the AMOC strength anomaly for all individual models in Figure 4c. This shows a clear relationship between the mPWP-PI change in density gradient and the AMOC strength, where a greater increase in density gradient results in a larger increase
in AMOC strength. A notable exception is GISS2.1G, in which deep water formation takes place at higher latitudes than 65°N and the AMOC strength is thus not strongly related to the density gradient we have defined.

Changes in density are brought about by changes in the potential temperature and salinity of the ocean. We show the potential temperature and salinity in the same manner as the potential density in Figure 4d-4f and Figure 4g-4i, respectively. As the North
Atlantic becomes substantially warmer between 40-80°N than the rest of the basin in the mid-Pliocene, temperature cannot explain the increase in meridional density gradient. This leaves salinity as the main driver of the increase in the meridional density gradient, which is supported by Figures 4g-4i. We see a zonal mean freshening of the South Atlantic and tropics from the surface down to approximately 500 m. The rest of the basin shows an increase in salinity, with the exception of the surface at latitudes higher than 65°N. A high salinity anomaly is found between 40-60°N in the top 100 m, extending further downwards
in some locations. The salinity anomaly results in an increase in the meridional salinity gradient in the mid-Pliocene, where a relationship between the increased salinity gradient and increased AMOC strength in the mid-Pliocene is shown in Figure 4i. Overall, Figure 4 shows that the density driven increase in AMOC strength must be a result of increased salinity in the high North Atlantic.



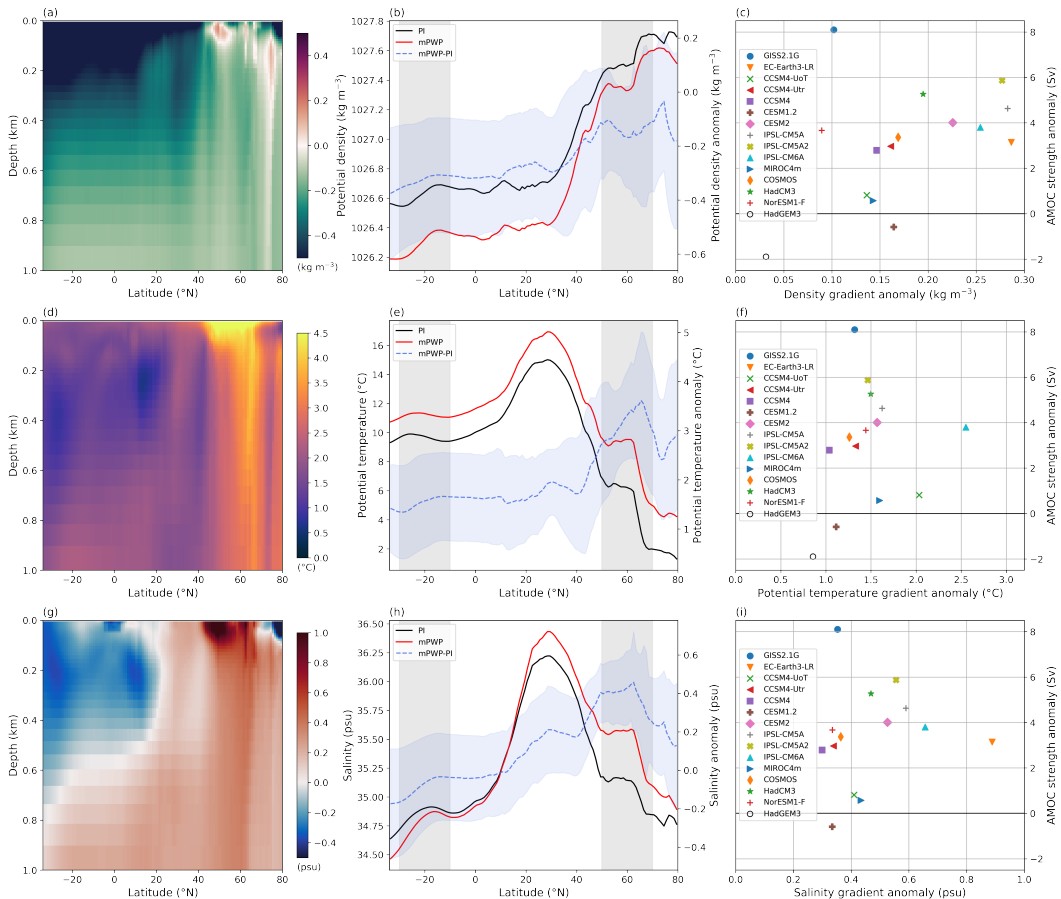

**Figure 4.** (a) MMM top 1 km Atlantic zonal mean potential density mPWP-PI anomaly. (b) MMM top 1 km depth-averaged Atlantic zonal mean potential density in the PI and mPWP (left y-axis) and mPWP-PI anomaly (right y-axis). Shading indicates one standard deviation from the MMM by individual models. (c) Individual model mPWP-PI anomaly in meridional gradient of the top 1 km Atlantic potential density, plotted against the mPWP-PI AMOC strength anomaly. The meridional gradient is defined as the difference between the 50-70°N average and 10-30°S average (latitude bands are indicated by grey shading in (b)). (d)-(f) Same as (a)-(c) for potential temperature. (g)-(i) Same as (a)-(c) for salinity.

### 3.2.1 Sea surface salinity

The consistent intensification of the mid-Pliocene AMOC across the PlioMIP2 ensemble compared to the PlioMIP1 ensemble is suggested to be linked to the closure of the Arctic gateways in PlioMIP2 (Haywood et al., 2020; Zhang et al., 2021b). Closing these Arctic gateways has been shown to cause an intensification of the AMOC through the decrease in freshwater transport from the Arctic into the North Atlantic via the Labrador Sea. This increases the salinity in the Labrador Sea and subpolar North Atlantic, thereby stimulating deep water formation in these areas (Otto-Bliesner et al., 2017). In the previous section we show

that there is a substantial increase in salinity in the 40-60°N latitude band in the North Atlantic. From the MMM sea surface

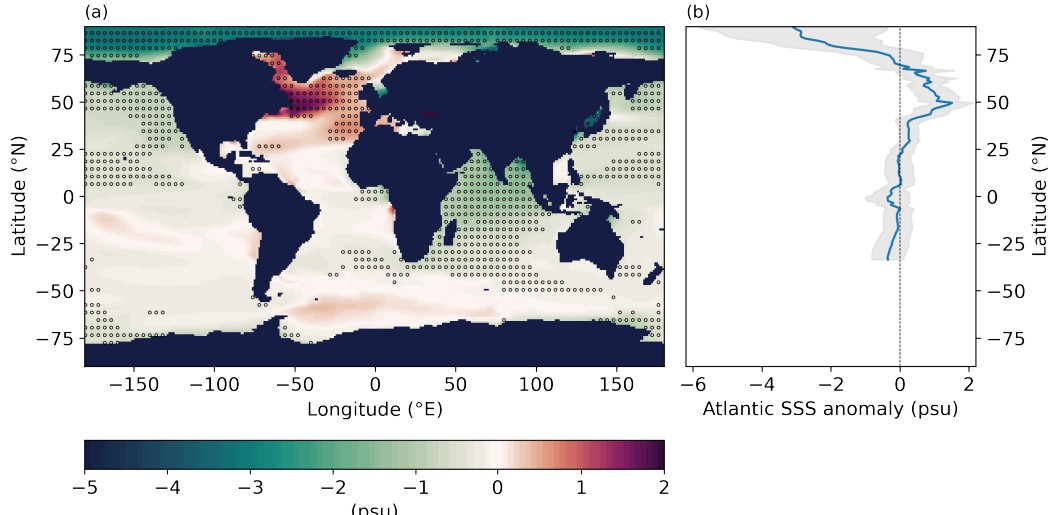

**Figure 5.** (a) Multi-model mean mPWP-PI difference in sea surface salinity (SSS). Stippling indicates that 12 or more models agree on the sign of the difference. (b) Atlantic zonal mean SSS anomaly, excluding the Mediterranean Sea and North Sea. The shading indicates one standard deviation from the MMM by individual models, excluding HadGEM3.

salinity (SSS) in Figure 5a, we can identify a robust increase of approximately 2 psu in sea surface salinity in the Labrador Sea and in the North Atlantic between 40-60°N. It is exclusively at these latitudes that we find that Atlantic zonal mean sea surface salinity anomaly to be significantly positive (Figure 5b) when considering the standard deviation of the individual models. All models forced with the PRISM4 reconstruction in the mid-Pliocene show this increase in sea surface salinity, as can be seen in

Supplementary Figure S1. The fact that the HadGEM3 model, using a pre-industrial land-sea mask, does not show a sea sur-face salinity increase in the subpolar North Atlantic or Labrador Sea strengthens the argument that the stronger mid-Pliocene AMOC can be attributed to the closure of the Arctic gateways in the PlioMIP2 simulations.

Even though the Bering Strait is closed in the mid-Pliocene experiments, blocking freshwater transport from the Pacific into

the Atlantic Ocean, a 3-4 psu decrease in sea surface salinity in the Arctic Ocean can be seen in Figure 5a. This decrease is present in all models except GISS2.1G but the extent and magnitude of the decrease is highly model dependent (Supplemen-tary Figure S1). While the closure of the Bering Strait could be expected to lead to higher sea surface salinity in the Arctic Ocean, we observe the opposite. This may be explained by earlier PlioMIP2 studies that show a higher surface freshwater flux into the Arctic Ocean in the mid-Pliocene (Haywood et al., 2020; Han et al., 2021) as a result of atmospheric warming. Other

possible factors include an increase in runoff due to the reduced Greenland Ice Sheet and a decrease in sea-ice extent in the mid-Pliocene (de Nooijer et al., 2020). In the next subsection, we will consider the role of the gateway closure in the freshwater transport between the Arctic, North Pacific and North Atlantic and how this contributes to the changes in sea surface salinity that we observe here.





### 3.2.2 Freshwater transport

In the pre-industrial, freshwater exchange of the Atlantic basin with the Pacific Ocean occurs both at 34°S and through the Bering Strait. The Pacific water that is transported to the Arctic through the Bering Strait is relatively fresh. As the Arctic freshwater is then transported to the Atlantic Ocean via the Canadian Archipelago and Fram Strait into the Labrador and Norwegian Seas, it dampens convection and deep water formation in the high North Atlantic. As mentioned in earlier sections, it has been shown that closing the Bering Strait and Canadian Archipelago decreases the freshwater transport into and out from the Arctic Ocean (Hu et al., 2015; Otto-Bliesner et al., 2017), promoting stronger deep water formation in the North Atlantic. In this section, we will take a closer look at both the freshwater transport through the whole Atlantic basin as well as the exchange of freshwater between the North Pacific and Arctic through the Bering Strait and between the Arctic and North Atlantic via the Fram Strait and Labrador Sea.

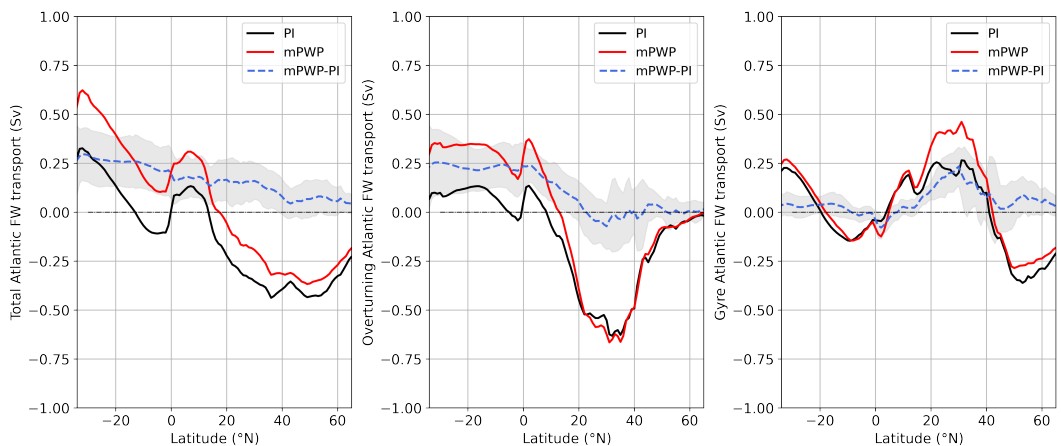

**Figure 6.** (a) MMM total Atlantic freshwater transport in the mid-Pliocene, pre-industrial and the difference. (b) MMM Atlantic freshwater transport by overturning. (c) MMM Atlantic freshwater transport by wind-driven gyres. The shading indicates one standard deviation of deviation from the MMM difference by individual models, excluding HadGEM3.

Figure 6a shows that the MMM southward total freshwater transport into the Atlantic Ocean at 65°N decreases by 0.04 Sv (-19%) in the mid-Pliocene. This is presumably the main cause of the higher MMM sea surface salinity in the North Atlantic shown in Figure 5. At 34°S, the southern boundary of the Atlantic Ocean, the MMM northward transport of freshwater increases by 0.27 Sv (+96%) in the mid-Pliocene. Using Figure 6b and 6c, we can determine whether changes we observe in the total freshwater transport can be attributed to changes in the overturning or gyre circulation. Looking at the freshwater imported at the southern Atlantic boundary in Figure 6b and 6c, it is clear that the higher mid-Pliocene import of freshwater at the southern boundary can be attributed almost entirely to the overturning circulation. However, in the North Atlantic the change in Atlantic freshwater transport by overturning is relatively small and inconsistent at latitudes higher than 20°N. At



latitudes above 20°N, differences in the total Atlantic freshwater transport between the mid-Pliocene and pre-industrial can primarily be linked to the wind-drive gyre circulation, as can be seen in Figure 6c. The freshwater transport by the (northern)

subtropical gyre is almost doubled in the mid-Pliocene, while the transport by the subpolar gyre is decreased at latitudes higher than 50°N.

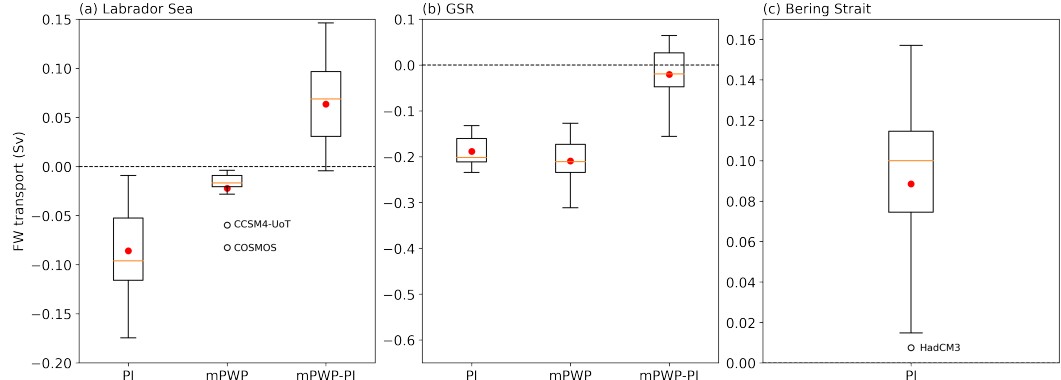

**Figure 7.** Boxplots of the mean freshwater transport (Sv) in pre-industrial, mid-Pliocene and their difference at 62°N (a) near the northern boundary of the Labrador Sea and (b) across the GSR. (c) Mean freshwater transport (Sv) through the Bering Strait in the pre-industrial. The box covers Q1 (25th percentile) to Q3 (75th percentile), with the median indicated by a horizontal orange line. The whiskers extend from Q1 to Q1-1.5*IQR and from Q3 to Q3+1.5*IQR where IQR=Q3-Q1. A model that falls outside the whiskers is separately shown as an unfilled circle. The MMM is indicated by a red filled circle.

To understand how the gateway closure impacts the freshwater transport from the Arctic into the Atlantic Ocean, we differentiate between the total freshwater transport on the western and eastern side of Greenland at 62°N in Figure 7a and 7b. The

freshwater transport on the western side of Greenland is entering the Labrador Sea and the transport on the eastern side is going across the Greenland Scotland Ridge (GSR). Note that in the mid-Pliocene, freshwater entering the Labrador Sea from the north cannot originate from the Arctic Ocean and can only result from runoff and the surface freshwater flux into Baffin Bay.

All model simulations show freshwater transport southwards into the Labrador Sea both in the pre-industrial and mid-Pliocene,

regardless of an open or closed Canadian Archipelago (see Supplementary Figure S3 for individual model results). However, the MMM freshwater transport into the Labrador Sea (Figure 7a) decreases dramatically with 0.063 Sv (-74%) from the pre-industrial to the mid-Pliocene. We do not observe this decrease in HadGEM3. In Figure 7b, we can see that the MMM freshwater transport increase of 0.021 Sv (+11%) across the GSR in the mid-Pliocene is relatively small and compensates a third of the decrease in freshwater transport into the Labrador Sea. In addition, the box plots indicate that the decrease in fresh-

water transport into the Labrador Sea is a more consistent feature among the models than the increase of freshwater transport across the GSR. These results are in line with the decrease of the MMM total freshwater transport from the Arctic Ocean to




the Atlantic Ocean in the mid-Pliocene (Figure 6a). The decrease in freshwater transport into the North Atlantic can partly be explained by the closure of the Bering Strait in the mid-Pliocene, causing a MMM decrease of 0.09 Sv freshwater transport into the Arctic (Figure 7c). It has also previously been shown that closing the Bering Strait leads to less freshwater transported

from the Arctic into the North Atlantic (Hu et al., 2015) due to effects on sea-ice motion and freshwater exchange.

While the decrease in freshwater transport plays a crucial role in the increased salinity in the Labrador Sea and subpolar North Atlantic, sea-ice extent and surface freshwater flux also play an important role in the Arctic freshwater balance (Aagaard and Carmack, 1989). It should be noted that the transport of sea-ice is not included in our freshwater transport calculations.

However, all models show a greatly reduced sea-ice cover in the Labrador Sea and Baffin Bay and in most models the annual sea-ice coverage does not extend to 62°N in the mid-Pliocene (Supplementary Figure S4). Therefore, qualitatively, our results will not be impacted if transport of sea-ice were to be taken into account. The role of changes in surface freshwater flux is explored further in the next subsection.

### 3.2.3 Surface freshwater flux

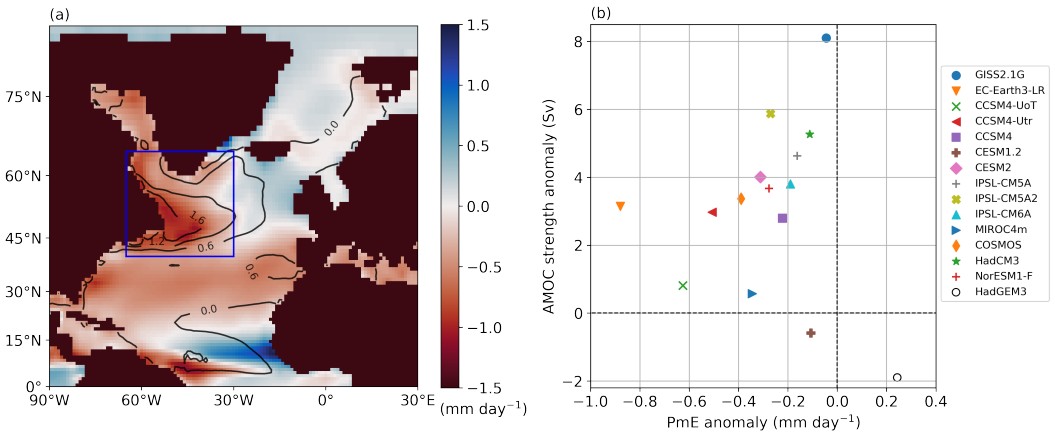

**Figure 8.** (a) Shading shows the MMM mPWP-PI difference in PmE freshwater flux. Black contours show the MMM mPWP-PI sea surface salinity anomaly. (b) Average mPWP-PI PmE anomaly (40-65°N, 30-70°W) plotted against the mPWP-PI AMOC strength anomaly. The area over which the PmE anomaly is averaged is indicated in (a) by the blue box. A negative PmE anomaly means that the evaporation is higher than precipitation and thus indicates saltening.

Figure 8 allows us to consider the role that the atmospheric freshwater flux plays in the increased salinity in the mid-Pliocene subpolar North Atlantic. The atmospheric freshwater flux is defined as the difference between the precipitation and evaporation (PmE), where a positive PmE means that precipitation exceeds evaporation at the surface. Shading in Figure 8a shows the MMM PmE anomaly, with the sea surface salinity anomaly indicated by the black contours. Off the coast of Newfoundland, the PmE shading follows sea surface salinity contours quite closely, which is an indication that the sea surface salinity is





influenced by changes in freshwater flux there. However, the PmE balance is negative over part of the subpolar North Atlantic experiencing a relatively high sea surface salinity increase of 0.6-1.2 psu. When plotting the PmE anomaly in the subpolar North Atlantic, averaged over the area indicated by the blue box in Figure 8a, against the AMOC strength anomaly for individual models in Figure 8b, we find no indication that the PmE freshwater flux is the driving force behind the mid-Pliocene AMOC strength increase. If this were the case, we would expect a more negative PmE anomaly to correspond to a stronger AMOC, as a negative PmE anomaly would cause higher salinity and thus higher ocean water density in the North Atlantic. Such a trend is not indicated by Figure 8b as linear regression analysis reveals no significant correlation ($R = -0.06$, $p = 0.83$).

### 3.3 Ocean heat transport

For all models with an intensified mid-Pliocene AMOC, we would expect the Atlantic OHT in the Northern Hemisphere to also be strengthened in the mid-Pliocene simulations. However, figure 9a shows that while the MMM total Atlantic OHT does increase at some latitudes, this increase is relatively small with a maximum of 0.09 PW (+9%) at 24 °N. Figure 9b and 9c show the relative contributions of the overturning and wind-driven gyre circulation to the MMM OHT. The MMM Atlantic OHT by overturning increases significantly more than the total Atlantic OHT in the NH with a maximum of 0.20 PW (+23%) at 24°N. Between 20-40°N, this increase is robust across the ensemble when considering the standard deviation.

While the heat transported by the overturning circulation increases in the mid-Pliocene North Atlantic Ocean, the MMM gyre Atlantic OHT decreases (see figure 9c). This decrease is strongest in the region of the subtropical gyre, with a maximum decrease of -0.12 PW (-15%) at 30°N and a decrease of -0.10 PW (-12%) at 24°N. This suggests that the gyre circulation responds to mid-Pliocene conditions in such a way that it compensates for approximately half of the increase in Atlantic OHT by overturning in the subtropical North Atlantic. When we consider only the latitudes of the subtropical North Atlantic, 20-40°N, we find that the 20-40°N average MMM total Atlantic OHT at these latitudes increases by 0.07 PW (+9%) and the overturning Atlantic OHT increases by more than twice this value: 0.16 PW (+21%).

Figure 10 shows the mPWP-PI change in AMOC strength and total and overturning Atlantic OHT for all models. The OHT is averaged over 20°N-40°N for three reasons: the influence of the overturning circulation on the Atlantic OHT is largest between 20°N-40°N, it is the region where the subtropical gyre also influences the OHT, and most mid-Pliocene simulations show amplified SST warming in the Atlantic Ocean above 40°N. When we compare changes in AMOC strength to those in the total OHT, not all models show that a stronger AMOC is accompanied by a higher total Atlantic OHT. However, when considering the OHT by overturning, all models with an intensified mid-Pliocene AMOC also have enhanced Atlantic OHT by overturning. A linear least-squares regression performed on the AMOC and overturning OHT reveals a slope that is 42% higher than slope of the regression between the AMOC and total OHT, as well as a better fit indicated by a higher $R^2$ and substantially lower p-value. This result indicates that OHT components should be considered separately when looking at the response of the Atlantic OHT to a stronger mid-Pliocene AMOC.



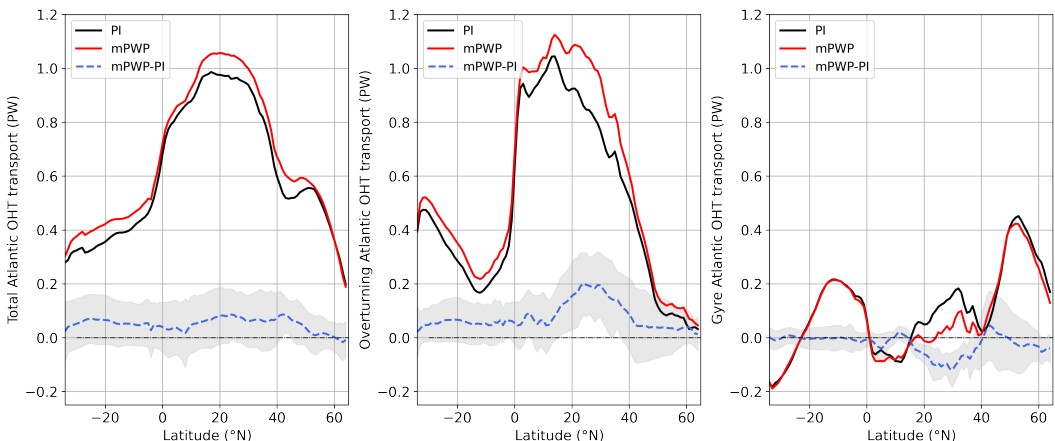

**Figure 9.** (a) MMM total Atlantic ocean heat transport in the mid-Pliocene, pre-industrial and the difference. (b) MMM Atlantic ocean heat transport by overturning. (c) MMM Atlantic ocean heat transport transport by wind-driven gyres. The shading indicates one standard deviation by individual models from the MMM.

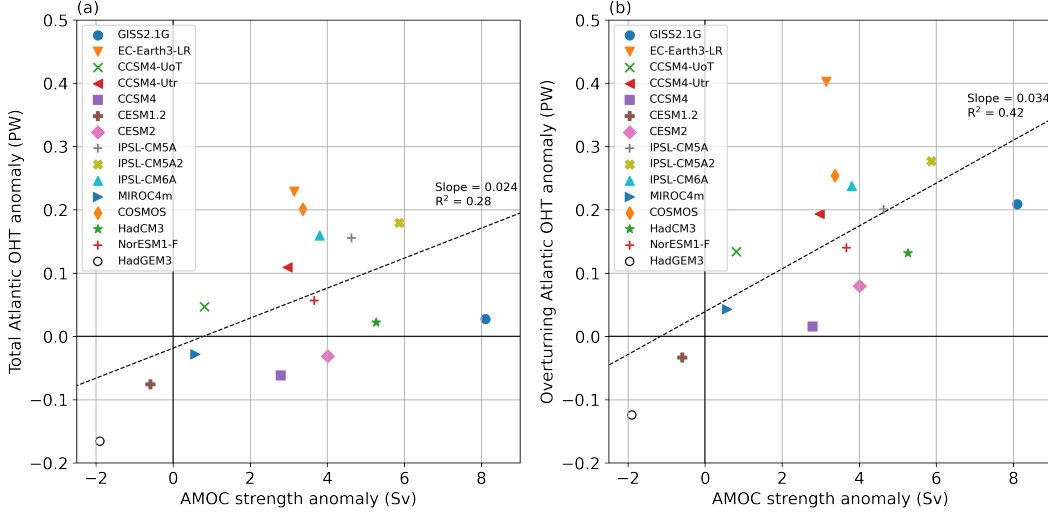

**Figure 10.** (a) mPWP-PI change in average total Atlantic OHT between 20°N-40°N and change in AMOC strength. (b) mPWP-PI change in overturning Atlantic OHT between 20°N-40°N and change in AMOC strength. Dotted line indicates linear fit by least squares with the slope and $R^2$ of fit shown.

In Table 2, the results of least-square linear fits between the total OHT and the AMOC strength and the overturning OHT and the AMOC strength are shown for different latitude ranges over which the OHT is averaged. When considering the majority of the Atlantic Ocean basin (30°S-60°N), the regression shows an increased slope and $R^2$ value when performing it on the overturning OHT component rather than on the total OHT. We have performed the regressions also when excluding EC-Earth3-



**Table 2.** Results of least square linear fit between the total or overturning OHT mPWP-PI anomaly with the AMOC strength anomaly. The latitude range indicates over what latitudes the OHT has been averaged.

| | OHT component | Latitude range OHT | Slope (PW/Sv) | Intercept (PW) | $R^2$ | P-value |
|---|---|---|---|---|---|---|
| All models | Total | 20°N-40°N | 0.023 | -0.019 | 0.28 | 0.044 |
| | Overturning | 20°N-40°N | 0.034 | 0.040 | 0.42 | 0.009 |
| | Total | 30°S-60°N | 0.018 | -0.014 | 0.25 | 0.056 |
| | Overturning | 30°S-60°N | 0.021 | 0.006 | 0.33 | 0.025 |
| All models excluding | Total | 20°N-40°N | 0.024 | -0.031 | 0.33 | 0.031 |
| EC-Earth3-LR | Overturning | 20°N-40°N | 0.034 | 0.021 | 0.59 | 0.001 |
| | Total | 30°S-60°N | 0.018 | -0.023 | 0.29 | 0.045 |
| | Overturning | 30°S-60°N | 0.020 | -0.006 | 0.47 | 0.006 |

LR, which is an outlier in Figure 9a and 9b due to its exceptionally large OHT increase. When EC-Earth3-LR is excluded, the $R^2$ increases significantly for all cases, especially for the regression with the overturning OHT, without substantial changes in the slope. In addition, the y-intercept is closer to zero and the p-value decreases dramatically for the overturning OHT. Overall, these results suggest that the overturning OHT is a better indicator of the direct response of the OHT to a stronger AMOC in the mid-Pliocene. Overall, the PlioMIP2 ensemble shows a rather consistent response in the Atlantic OHT associated with overturning, where a stronger AMOC leads to enhanced Atlantic OHT by overturning.

### 3.4 Transports by the subtropical gyre

Figure 9c showed that the MMM Atlantic OHT by the wind-driven gyre circulation shows a substantial decrease in the (northern) subtropical gyre region. At the same time, the MMM Atlantic freshwater transport of the northern subtropical gyre is almost doubled in the mid-Pliocene (Figure 6c), meaning that less salt is being transported northwards by the subtropical gyre circulation. One possibility is that the subtropical gyre circulation itself responds to mid-Pliocene conditions through its coupling to the atmosphere. As wind stress fields are not available for the entire PlioMIP2 ensemble, we investigate changes in surface winds that drive the gyre circulation in Figure 11. Figure 11a shows the zonal mean curl of the MMM wind velocity at 1000 hPa in the North Atlantic. A positive wind velocity curl is associated with counterclockwise motion, the subpolar gyre, and a negative wind velocity curl with clockwise motion, the subtropical gyre. The wind velocity curl in the subpolar region shows a substantial decrease, indicating weakening of the subpolar gyre circulation. In the subtropical gyre region, there is a shift in the wind velocity curl minimum towards higher latitudes but we do not see a significant change in magnitude of the MMM zonal mean wind velocity curl. This result indicates that there are no substantial changes in the mid-Pliocene wind circulation driving the subtropical gyre, and thus not in the subtropical gyre circulation itself. The lack of change in the subtropical gyre circulation is confirmed by the upper 500 m ocean meridional velocity, averaged over the 20-40°N latitude band, which is similar in the mid-Pliocene and pre-industrial outside of the highly variable Gulf Stream region (Supplementary Figure S6).



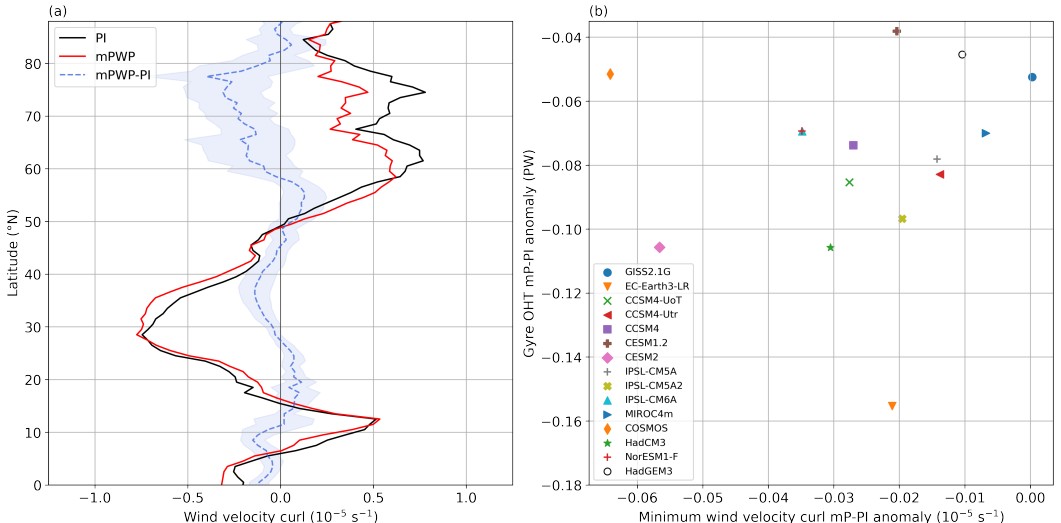

**Figure 11.** (a) MMM Atlantic zonal mean 1000 hPa wind velocity curl ($\frac{dv}{dx} - \frac{du}{dy}$) in the pre-industrial and mid-Pliocene, and their difference. Shading indicates one standard deviation from the MMM difference by individual models. (b) The minimum zonal mean 1000 hPa wind velocity curl between 20-40°N plotted against the 20-40°N average OHT gyre component anomaly for individual models.

Figure 11b also does not suggest a connection between changes in the surface winds and the OHT by the subtropical gyre. No

consistent relationship can be found between the minimum wind velocity curl in the subtropical gyre region and Atlantic OHT by the subtropical gyre when performing a least-squares linear regression ($R = 0.11$, $p = 0.69$). Therefore, we do not see any changes in the subtropical gyre circulation that could explain the substantial decrease in OHT by the gyre between 20-40°N

Without substantial changes to the wind-driven gyre circulation itself, changes in OHT and freshwater transport by the sub-

tropical gyre may be linked to zonal asymmetry in the mPWP-PI anomalies in temperature and salinity. A zonal asymmetry in temperature or salinity could lead to more transport on the eastern or western part of the Atlantic basin, influencing the total gyre transport. Figure 12a and Figure 12b show the difference between mid-Pliocene and pre-industrial SST and SSS respectively for the subtropical gyre region. A zonal asymmetry can be seen in both fields, with the eastern Atlantic becoming relatively warmer and saltier than the western Atlantic in the mid-Pliocene. When taking the meridional average over 20-40°N

for the SST and SSS, we find a zonal gradient in SST and SSS anomalies in Figure 12c and Figure 12d. For each model, the change in zonal SST gradient averaged between -20 and -70°E (shaded grey in Figure 12c) is plotted against the Atlantic gyre OHT anomaly in Figure 12e. This reveals a significant negative correlation ($R = -0.79$, $p < 0.05$) meaning that a larger zonal gradient in SST anomalies corresponds to a larger decrease in OHT by the gyre. The same plot is shown for the average zonal gradient in the SSS anomalies and the Atlantic gyre freshwater transport in Figure 12f. While the relationship is less robust

than that between the SST gradient and OHT, a significant positive correlation ($R = 0.66$, $p < 0.05$) is found where a larger gradient in SSS anomalies is related to more northward freshwater transport by the subtropical gyre.

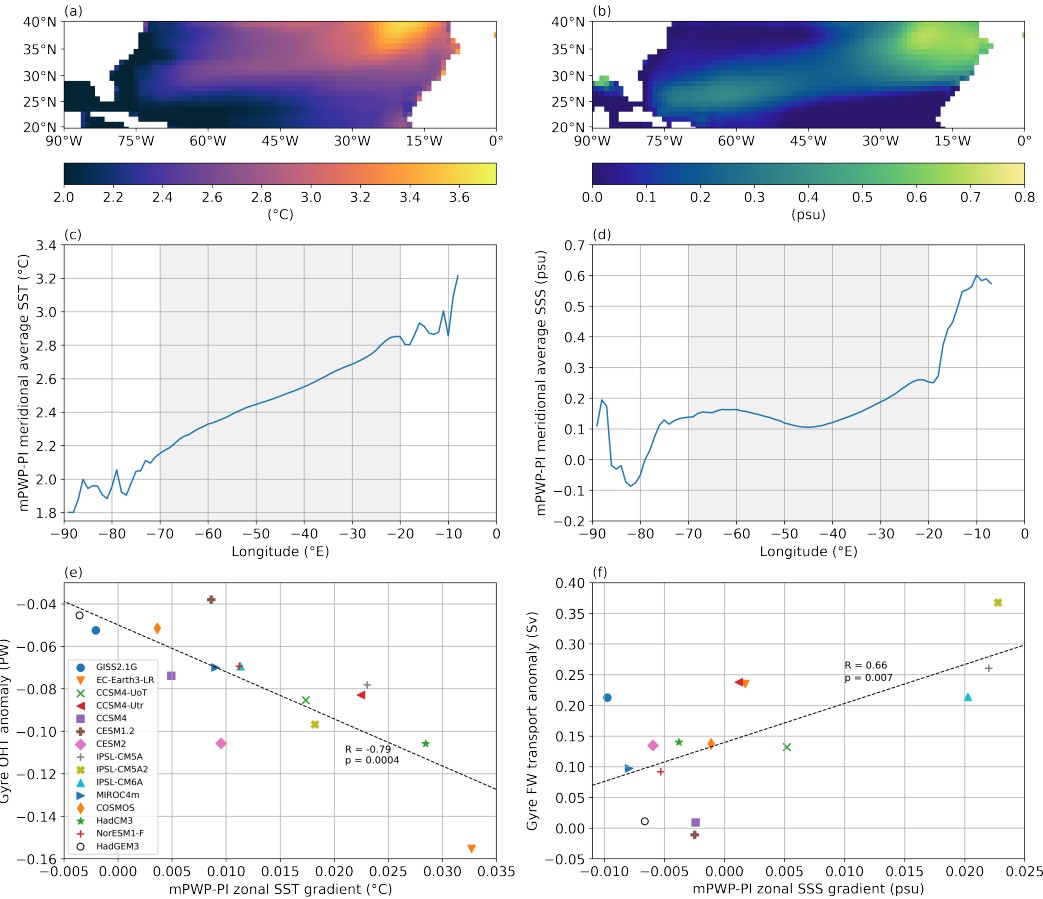

**Figure 12.** (a) MMM mPWP-PI Atlantic SST difference between 20-40°N. (b) MMM mPWP-PI Atlantic SSS difference between 20-40°N. (c) MMM meridional mean mPWP-PI Atlantic SST difference between 20-40°N. (d) MMM meridional mean mPWP-PI Atlantic SSS difference between 20-40°N. (e) Individual model mPWP-PI mean difference in zonal SST gradient from (c) between 20-70°W plotted against the mPWP-PI difference in OHT gyre component. (f) Individual model mPWP-PI mean difference in zonal SSS gradient from (d) between 20-70°W plotted against the mPWP-PI difference of the gyre freshwater transport.

## 4 Discussion

While there is no consistent relationship between the AMOC strength and average North Atlantic SST warming (Zhang et al., 2021b), the correlation maps (Figure 3) illustrate that a stronger mid-Pliocene AMOC exerts a relatively larger influence on

North Atlantic SSTs. A subset of the models does not show an increase in correlation in the mid-Pliocene: CCSM4, CESM1.2, CESM2, IPSL-CM6A and HadGEM3. As HadGEM3 uses the same land-sea mask in their mid-Pliocene simulations, the lack of stronger correlation is expected for this model. For the other models, the reason is not so evident. Three out of these four models, CCSM4, CESM1.2 and CESM2, come from the same model family. CCSM4 and CESM2 also show a relatively small





increase in overturning OHT with a stronger mid-Pliocene AMOC compared to the other models, even though their sea surface
salinity and gateway transport are in line with ensemble behavior.

The high correlation found between North Atlantic SSTs and the AMOC strength in the mid-Pliocene suggests that there
likely is a relationship between the stronger AMOC and the amplified North Atlantic SST warming in the mid-Pliocene in the
PlioMIP2 ensemble. However, the effect of the AMOC on North Atlantic SSTs is strongly model dependent and cannot be
simply quantified. Other important factors that may influence the SSTs are atmospheric processes and the presence of sea-ice
(Zhang et al., 2019), as well as model dynamics. For instance, enhanced vertical mixing has been shown to promote surface
temperature anomalies at high latitudes in the Pliocene (Lohmann et al., 2022). These factors vary among the different models
(e.g. Supplementary Figure S4) and cause a varied response of North Atlantic SSTs to mid-Pliocene boundary conditions and
$CO_2$ increase. However, Burton et al. (in prep.) show that North Atlantic SSTs in the mid-Pliocene are predominantly driven by
changes in geography and ice sheets. As we have shown that the mid-Pliocene AMOC strengthens as a result of these boundary
conditions, specifically the closure of the Arctic gateways, the amplified warming of North Atlantic SSTs and the intensified
AMOC are related.

When considering proxy SST data in the North Atlantic, sites 609, 982, 642 and 907 are specifically of interest as they are clos-
est to above-average MMM SST warming in the North Atlantic (Figure 2b). Disregarding HadGEM3, there are four models
at these sites that consistently show the highest warming: EC-Earth3-LR, CCSM4-Utr, CCSM4-UoT and CESM2 (Figure 2c).
While these models all have a stronger mid-Pliocene AMOC, they do not have another common factor that directly connects
this warming to a stronger mid-Pliocene AMOC, such as a large increase in overturning or total OHT or a strong correlation
of the AMOC to North Atlantic SSTs. It has however been shown by Haywood et al. (2020) that these four models have the
highest Earth system sensitivity (ESS) of the PlioMIP2 ensemble together with CESM1.2. On the other hand, their equilibrium
climate sensitivities (ECS) vary, where only CESM2 and CESM1.2 are among the highest in the ensemble. As the ESS takes
into account long-term feedbacks from ice sheets and paleogeographic boundary conditions, the relatively high warming in
these models, which aligns with the warm reconstructed SSTs at site 962 and 942, may be related to these feedbacks. This
strengthens the argument that the AMOC likely does play a role in the amplified North Atlantic warming, as the stronger
AMOC is a direct consequence of the boundary conditions. HadGEM3 showing high warming at sites 642 and 907 is probably
related to the fact that its ECS of 5.55°C (Andrews et al., 2019) is the highest of all PlioMIP2 models. Note that HadGEM3 is
not among the warmest models at sites 609 and 982, which we expect are the sites that are impacted most by increased OHT
due to a stronger mid-Pliocene AMOC.

The AMOC becomes stronger in almost all mid-Pliocene simulations, despite the above-average warming in the subpolar
North Atlantic. In future simulations, ocean warming often leads to a weakening AMOC by inhibiting deepwater formation.
In our results we show that the strengthening of the AMOC in the mid-Pliocene is a result of high salinity in the Labrador Sea
and subpolar North Atlantic. The high salinity is principally caused by decreased freshwater transport into the Labrador Sea



due to the closure of the Canadian Archipelago. While our results show that a negative Atlantic PmE anomaly plays a role by
possibly amplifying the increase in salinity in the mid-Pliocene, they do not suggest that the PmE anomaly is the main cause
of the high salinity driving the stronger mid-Pliocene AMOC. Another factor in the increased salinity could be the absence of
sea-ice over the Labrador Sea in the mid-Pliocene (Supplementary Figure S4). However, we find that the annual sea-ice cover
does not extend into the Labrador Sea in the pre-industrial for the majority of the models. In addition, the high salinity persists
below the sea surface as shown by the average top 100 m salinity (Supplementary Figure S2). This field shows the same pattern
as the sea surface salinity and its mPWP-PI anomaly is of comparable magnitude. Overall, these results support that the high
mid-Pliocene (sea surface) salinity in the Labrador Sea and subpolar North Atlantic is a result of the closure of the Bering
Strait and Canadian Archipelago.

The model response of the total Atlantic ocean heat transport (OHT) to a stronger mid-Pliocene AMOC is diverse: not all
PlioMIP2 models that simulate an intensified AMOC also show enhanced Atlantic OHT in the mid-Pliocene. The opposing
response of OHT by overturning and OHT by the gyre in the subtropical gyre region is very likely to be an important cause
of this diversity, where individual model dynamics govern the degree to which the two components compensate each other's
increase or decrease. However, when considering the overturning component only, all of the models that have a stronger mid-
Pliocene AMOC show stronger associated OHT.


We expected that the opposing response of the subtropical gyre OHT component would result from coupling between the
gyre circulation and the atmosphere. In that case, an increase in AMOC strength would lead to increased Atlantic OHT by
overturning between 20°N-40°N. This higher OHT by overturning decreases the meridional temperature gradient between
these latitudes in the ocean and thereby also at the surface. A decrease in surface winds due to the reduced meridional temper-
ature gradient at the surface would induce a weakening in the subtropical gyre circulation and thus a decrease in strength of
the Atlantic OHT gyre component. While this seems to be a plausible mechanism for explaining the opposite behavior of the
overturning and gyre OHT component, our results do not show that this mechanism plays role in the response of the PlioMIP2
ensemble as there are no substantial changes in the gyre circulation itself. It appears that changes in the ocean temperature and
salinity fields are responsible for altering the OHT and freshwater transport by the subtropical gyre. We observe that between
20°N and 40°N, the eastern part of the Atlantic basin becomes relatively warmer and saltier than the western part in the mid-
Pliocene. This results in the gyre circulation transporting relatively more heat and salt southwards than in the pre-industrial.
The relatively high temperature and salinity in the eastern subtropical North Atlantic appears to originate from the even warmer
and saltier mid-Pliocene subpolar North Atlantic, features that we have related to the stronger AMOC and Arctic gateway clo-
sure, respectively. The warm and salty water in the subpolar North Atlantic is transported eastwards by the northern branch
of the subtropical gyre and subsequently transported southwards, resulting in the zonal asymmetry we observe. It is possible
that the weakened subpolar gyre circulation plays a role in increased eastward transport of the warm and salty subpolar North
Atlantic water.



Our results point to a mechanism of compensation between the OHT components in the subtropical gyre region in the Atlantic Ocean. This compensation is, however, not via the atmosphere as has been suggested previously to be the case in the pre-industrial (Vallis and Farneti, 2009; Farneti and Vallis, 2013). Rather, our results for the mid-Pliocene indicate that the enhanced warming in the mid-Pliocene subpolar North Atlantic SSTs causes more eastward and subsequent southward heat transport by the subtropical gyre. This suggests that the mid-Pliocene background state may affect OHT dynamics, and thereby highlights potential differences between the mid-Pliocene and near-future climate. We do not observe such a compensating mechanism in the overturning and gyre components of the freshwater transport.

## 5   Conclusions

In our study, we have employed fifteen models from the PlioMIP2 ensemble to investigate what drives the stronger AMOC in the mid-Pliocene simulations and how the Atlantic OHT and North Atlantic SSTs respond to this strengthening. All models that simulate a stronger mid-Pliocene AMOC show that the closure of the Bering Strait and Canadian Archipelago in the mid-Pliocene leads to a dramatic decrease in southward freshwater transport into the Labrador Sea. The ensuing increased salinity in the Labrador Sea and subpolar North Atlantic stimulates deepwater formation in these areas, leading to a stronger AMOC. These results agree with the conclusions of Otto-Bliesner et al. (2017), who showed that closing the Bering Strait and Canadian Archipelago in their CCSM4 model led to a stronger mid-Pliocene AMOC due to altered freshwater transport in the Arctic and North Atlantic. We have shown in this study that this is a consistent response to the closure of these Arctic gateways across the PlioMIP2 ensemble and is thereby the driver of the reported stronger AMOC in the mid-Pliocene simulations.

We show that the response of the total Atlantic OHT to a stronger mid-Pliocene AMOC seems inconsistent among models due to a compensation mechanism between the overturning circulation and wind-driven gyre circulation. When separating the OHT into two components, one driven by the overturning circulation and the other by the wind-driven gyre circulation, we find that the OHT associated with overturning does consistently increase with a stronger AMOC. The OHT associated with the subtropical gyre decreases as a response to ocean temperatures increasing more in the east than in the west of the North Atlantic in the mid-Pliocene. This decrease in gyre OHT partially compensates the higher OHT by overturning in the northern subtropical gyre region. As individual model dynamics are highly variable, the degree of compensation differs among models. However, the mean response is consistent. We argue that the OHT components should be considered separately when evaluating the response of the OHT to a stronger mid-Pliocene AMOC. When doing so, we find that the stronger mid-Pliocene AMOC in the PlioMIP2 ensemble does lead to enhanced Atlantic OHT by the overturning circulation.

It has been suggested that the decrease in data-model mismatch in the North Atlantic may be due to a stronger AMOC transporting more heat to the North Atlantic in the PlioMIP2 ensemble (Haywood et al., 2020). Indeed, our results show that the AMOC has a stronger influence on North Atlantic SSTs in the mid-Pliocene than in the pre-industrial. However, the spatial extent and magnitude of this effect is highly variable among individual models, which may explain why Zhang et al. (2021b)

were not able to identify a consistent relationship between the AMOC strength and average North Atlantic SST warming. It remains difficult to quantify the extent of the influence of the AMOC on North Atlantic SSTs, but our study shows that its influence is significant. Furthermore, we conclude that the AMOC is an important factor in explaining the better agreement of
the PlioMIP2 ensemble SSTs with reconstructions in the North Atlantic.

The results presented in this study provide an in-depth look at how the stronger mid-Pliocene AMOC in PlioMIP2 is driven and what its consequences are for the OHT and SST warming in the North Atlantic. They also raise further questions, such as the degree to which the decrease in OHT by the gyre circulation is a response to the increase in OHT by the stronger At-
lantic overturning circulation and how the ECS and ESS of individual models may influence modelled SST warming in the mid-Pliocene. Furthermore, given the influence of the stronger mid-Pliocene AMOC on the Atlantic OHT and enhanced SST warming, it raises the question as to what extent and in which context the mid-Pliocene is suitable as a future climate analog. The impact of a stronger AMOC on the regional and global climate is significant, and must be taken into consideration when comparing the mid-Pliocene climate to future warming scenarios.

*Code and data availability.* PlioMIP2 data used for this paper is available upon request from Alan M. Haywood (a.m.haywood@leeds.ac.uk), with the exception of IPSL-CM6A, EC-Earth3-LR and GISS2.1G. PlioMIP2 data from IPSL-CM6A, EC-Earth3-LR and GISS2.1G can be obtained from the Earth System Grid Federation (ESGF) (https://esgf-node.llnl.gov/search/cmip6/, last access: 14 February 2022). The $U_{37}^{k'}$ and Mg/Ca SST reconstructions from McClymont et al. (2020) can be obtained through https://doi.pangaea.de/10.1594/PANGAEA.911847 (last access: 24 January 2022) and the $U_{37}^{k'}$ SST reconstructions from Foley and Dowsett (2019) can be obtained through https://doi.org/10.5066/P9YP3DTV
(last access: 24 January 2022). The observational pre-industrial SSTs from the NOAA ERSST5 dataset (Huang et al., 2017) can be down-loade from https://www.ncei.noaa.gov/products/extended-reconstructed-sst (last access: 24 January 2022).

The Jupyter Notebooks used for data analysis are available through https://github.com/jweiffenbach/PlioMIP2-AMOC.

*Author contributions.* JEW, MLJB, HAD and ASvdH designed the work. JEW performed the analysis and wrote the manuscript of the paper.
The remaining authors provided the PlioMIP2 experiments and contributed to the discussion of the results and the contents of the manuscript.

*Competing interests.* The authors declare that they have no conflict of interest.

*Acknowledgements.* The work by Julia E. Weiffenbach, Michiel L. J. Baatsen, Henk A. Dijkstra and Anna S. von der Heydt was carried out under the program of the Netherlands Earth System Science Centre (NESSC), financially supported by the Ministry of Education, Culture and Science (OCW grant number 024.002.001)). CCSM4-Utr simulations were performed at the SURFsara Dutch national computing facilities



and were sponsored by NWO-EW (Netherlands Organisation for Scientific Research, Exact Sciences) under the projects 17189 and 2020.022.

Bette L. Otto-Bliesner, Esther C. Brady and Ran Feng acknowledge support from U.S. National Science Foundation grant numbers 1814029, and 1852977 (B.L.O and E.C.B). The CCSM4 and CESM1 and CESM2 simulations are performed with high-performance computing support from Cheyenne (doi:10.5065/D6RX99HX) provided by NCAR's Computational and Information Systems Laboratory, sponsored by the
National Science Foundation.

Alan M. Haywood and Julia C. Tindall acknowledge the FP7 Ideas programme from the European Research Council (grant no. PLIO-ESS, 278636), the Past Earth Network (EPSRC grant no. EP/M008.363/1) and the University of Leeds Advanced Research Computing service. Julia C. Tindall was also supported through the Centre for Environmental Modelling and Computation (CEMAC), University of Leeds.
Christian Stepanek acknowledges funding from the Helmholtz Climate Initiative REKLIM. Christian Stepanek and Gerrit Lohmann acknowledge funding via the Alfred Wegener Institute's research programme Marine, Coastal and Polar Systems.

W. Richard Peltier and Deepak Chandan were supported by Canadian NSERC Discovery Grant A9627, and they wish to acknowledge
the support of SciNet HPC Consortium for providing computing facilities. SciNet is funded by the Canada Foundation for Innovation under the auspices of Compute Canada, the Government of Ontario, the Ontario Research Fund – Research Excellence, and the University of Toronto.

Zhongshi Zhang and Xiangyu Li acknowledge financial support from the National Natural Science Foundation of China (grant no. 42005042),
the China Scholarship Council (201804910023) and the China Postdoctoral Science Foundation (project no. 2015M581154). The NorESM simulations benefitted from resources provided by UNINETT Sigma2 – the National Infrastructure for High Performance Computing and Data Storage in Norway.

Charles J. R. Williams and Dan Lunt acknowledge the financial support of the UK Natural Environment Research Council (NERC)-funded
SWEET project (research grant no. NE/P01903X/1), as well as the European Research Council under the European Union's Seventh Framework Programme (FP/2007-868 2013) (ERC grant agreement no. 340923).

PlioMIP simulations with GISS2.1G were made possible by the NASA High-End Computing (HEC) Program through the NASA Center for Climate Simulation (NCCS) at Goddard Space Flight Center.
Ning Tan, Camille Contoux and Gilles Ramstein were granted access to the HPC resources of TGCC under the allocations 2016-A0030107732, 2017-R0040110492 and 2018-R0040110492 (gencmip6) and 2019-A0050102212 (gen2212) provided by GENCI. The IPSL-CM6 team of the IPSL Climate Modelling Centre (https://cmc.ipsl.fr/, last access: 28 April 2021) is acknowledged for having developed, tested, evaluated and tuned the IPSL climate model, as well as having performed and published the CMIP6 experiments.
Qiong Zhang acknowledges support from the Swedish Research Council (2013-06476 and 2017-04232). The EC-Earth3-LR simulations were performed on resources provided by the Swedish National Infrastructure for Computing (SNIC) at the National Supercomputer Centre





(NSC) partially funded by the Swedish Research Council through grant No. 2018-05913.

Wing-Le Chan and Ayao Abe-Ouchi acknowledge funding from JSPS KAKENHI (Grant no. 17H06104) and MEXT KAKENHI (Grant no. 17H06323), and are grateful to JAMSTEC for use of the Earth Simulator.





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
