# Peer review of "Unraveling the mechanisms and implications of a stronger mid-Pliocene AMOC in PlioMIP2"

_Climate of the Past, 2022_

## Author Comment (AC1)

*AC: The authors would like to thank Anonymous Referee #1 for the detailed feedback and comments. We appreciate the time and effort that has gone into this extensive review of our manuscript, and we think that there are various comments and points raised that will improve our manuscript. We will answer the comments and questions below and indicate the changes we plan to make in the revised manuscript.*

This is a review of the submission "Unraveling the mechanisms and implications of a stronger mid-Pliocene AMOC in PlioMIP2" of Weiffenbach et al. to Climate of the Past. The paper is a welcome contribution that utilize CMIP/PMIP output beyond reporting, to make progress in narrowing the yet substantial gap in our understanding of the drivers and impacts of the AMOC in the present and the past. The paper is in general well-written in terms of language and readability. The papers has some interesting conclusions. It shows how the closing of the Bering Strait results in lower freshwater input to the Atlantic from the Arctic and how that results in a region of higher salinity in the subpolar North Atlantic, which could very well be the driver of the stronger AMOC changes. And it shows that the stronger overturning leads to higher heat transport, but that a decreased zonal temperature gradient means the gyre heat transport is decreased. The resulting impact on North Atlantic SST is complex and model dependent. Despite these and other insights, I have some concerns about the validity of some of these analysis and conclusions that may require a substantial revision.

**General comments:**

1) Splitting of Overturning and Gyre transports:

The overturning heat transport (OHT) and wind-driven gyre heat transport (GHT) calculations are approximated by the heat transport from the zonal mean flow (ZHT) and a-zonal flow (AZHT), respectively. The difference between the GHT and the AZHT and the OHT and ZHT are not discussed even though it could be substantial. In fact, they are presented as the same thing (e.g., L329 and elsewhere). At 20-40N the ocean interior (outside the Western Boundary, WB) is very close to Sverdrup balance. The anomalous meridional transport due to the overturning occurs in the WB. This is well known from theory, models, and also clear in Figure S6. The ZHT uses the zonal mean temperature and not the WB temperate and thus likely under-estimates OHT. Similarly, in the deep ocean we know there is no wind-driven gyre at these latitudes, so there should be no GHT. Yet, the mean WB flow is subtracted everywhere in the deep ocean to give a resultant AZHT, that can obviously not be wind-driven and should not be presented as GHT, at least not without an analysis and discussion of the error made in this. Similarly, the deep eastern interior temperature is probably not relevant to OHT in reality, but for ZHT it dominates since most of the basin is away from WB. The same would apply for freshwater transports. Given that the Miocene-PI temperature changes do not have the same spatial patterns in the models, with some warming more in the east or west than others, this error made in the mPWP-PI calculations by equating ZHT to OHT and AZHT to GHT would be different in the different models.

A more relevant way to calculate the GTH and OHT at these sub-tropical gyre latitudes may be as follows: Determine an estimate of the level-of-no-motion (where the interior velocity nears zero). Next, at each latitude, determine a longitude for the western boundary of the wind-driven gyre by integrating meridional transport from the east to the west, until the longitude where the total meridional transport from east to west is zero. Now calculate the actual HT from v and T in this region above the level-of-no-motion and east of the western boundary of the gyre and this should more accurately represent the heat transport from the wind-driven circulation than the AZHT estimate. The OHT is then the actual HT in the rest of the cross-section. The same goes for the freshwater transport. This calculation is not as easily transferrable to other latitudes.

However, most of the conclusions from the overturning and gyre heat and freshwater transports are drawn from 20-40N so this should work well. It is still an approximation of course, but should be much closer to the actual OHT and GHT.

The major conclusion in the paper that the GHT is weaker during the Pliocene while the OHT is stronger, should still hold, since it can be deduced directly from the fact that the wind-driven volume transports are constant (Fig S6) and the zonal temperature gradient is decreased.

*AC: The method used in our paper is a conventional method that separates heat/freshwater transport by meridional overturning circulations from horizontal circulations, introduced by Hall and Bryden (1982). It has been used in many studies of both observational (e.g. Johns et al., 2010) and model data (e.g. Viebahn et al., 2016, Jüling et al., 2021). That said, it is important to recognize that it does not fully separate the transport by the wind-driven gyre circulation and overturning circulation for reasons that are raised in the comment above, and we will address this more elaborately in our revised manuscript.*

*Using CCSM4-Utrecht, the model that the contact author is most experienced with, we have assessed whether separating out the western boundary current and the deep ocean (below 1100 m) and then adding the transports of the western boundary current and deep ocean to the overturning component changes the results of our original component separation. We find that the effect is minimal, leading to a ~0.01 PW increase/decrease in overturning/gyre components. We note that performing the adapted method does create artifacts in the OHT components at some latitudes due to errors that result from defining the edge of the western boundary current. These results support that, even though our method is imperfect, it is robust for separating the overturning and gyre OHT components. Still, we will address the error associated with our current method in the methods and discussion section of our revised manuscript.*

*We have also tried to use the method suggested by the reviewer on CCSM4-Utrecht. We find that doing so is difficult due to large variations within the meridional velocity field at different latitudes. These variations are especially problematic for reliably determining the level of no motion. Without being able to accurately determine this level of no motion, the method resulted in completely unrealistic transports. An additional complication is that the meridional velocity fields also vary largely among models. We find that it would be a study in of itself to implement the suggested method for the whole PlioMIP2 ensemble and is from our point of view certainly beyond the scope of this study in terms of feasibility. Yet, we do acknowledge that this method should be further explored in future work and we will include it as a suggestion in our discussion.*

2) Cross-model correlations between quantities..

The figures Fig 4, Fig 8, Fig 10, Fig 11, Fig 12 are scatter plots involving all models to show the relationship of one process against another. The correlation coefficients and confidence levels are only given for Figure 8, 10, 12. It would be good to have in all these figures, rather than just a few. It is actually unlikely that one would find great correlations in these kinds of plots, even when processes are well correlated within one model. The constant of proportionality is usually highly model dependent. However, when two processes A and B both increase in (almost) all models at the Pliocene or both decreases, it is reasonable to assume that they are positively correlated somehow. A weak correlation coefficient between A and B across models, does not necessarily negate this. Of course, when the correlation is very strong, it does support the robustness of such a relationship within and among models. In this paper, the same standard for when processes are assumed to be related or to drive each other should be applied throughout for all these figures. At the moment this is not always the case. For instance, Fig 4c and

4i is said to show that the AMOC is driven by meridional and in particular salinity gradients. But the correlation coefficients are not given. Similarly, when all models show enhanced Arctic freshwater input to the Atlantic and a stronger AMOC, the freshwater decrease is taken to be the proven driven of the AMOC, even without correlation analysis. But in Figure 8 the near-zero correlation between PmE and the AMOC is taken to mean that PmE does not drive the AMOC directly. However, 14 of 15 models show more evaporation and a stronger AMOC.

*AC: We agree that the same standard should be applied for all analyses where there is an assumed relationship between processes. We will add the correlation coefficients and p-values to both Figure 4 and Figure 11, and we will refer to these in the text. In addition, we will make sure that these metrics are shown in the figures themselves for Figure 8 and Figure 10.*

*We also recognize that our arguments about the relationship between freshwater input from the Arctic and the strengthened AMOC should be supported by the same correlation analysis as used for the argument that PmE does not drive the AMOC directly. Preliminary further analysis shows that we may have underestimated the influence of the negative PmE anomaly on the North Atlantic salinity. We will carry out further correlation analyses and elaborate on the respective effects of the altered Arctic freshwater influx and of PmE in the mid-Pliocene on the North Atlantic salinity in our revised manuscript. We do not expect that there will be implications for the main findings of the manuscript, but we will nuance our earlier conclusion that a decrease in freshwater transport to the North Atlantic is the only driver of the stronger mid-Pliocene AMOC.*

3) Structure

The results at the moment jumps from 3.1 AMOC SST impact to 3.2 AMOC salinity drivers and then back to 3.3 AMOC heat transport impact. It seems more logical to keep the impact on SST and heat transport together. Perhaps start with the exploration of the salinity changes driving the AMOC changes, then continue to impacts.

*AC: We have tried out several structures while writing the manuscript and have chosen this one because it starts by addressing the main motivation behind our interest in the mid-Pliocene AMOC, which is to possibly explain the observed high SSTs in the mid-Pliocene. This relationship has been suggested in earlier literature based on SST proxy data as well as PlioMIP2 modelling data (e.g. Dowsett et al., 2013; Haywood et al., 2020) and links our manuscript to the greater body of papers about the Pliocene climate. Section 3.1, that addresses the relatively warm North Atlantic SSTs in the mid-Pliocene, provides this link and sets the stage for further analyses into why the AMOC is stronger (section 3.2) and whether associated changes in ocean heat transport would indeed support a relationship between warmer North Atlantic SSTs and a stronger AMOC in the Pliocene (section 3.3). Rather than changing the structure, we will elaborate on our motivation for this structure in the introduction to provide to the reader a clear guide through our arguments.*

**Specific comments:**

L68: Missing "is".

*AC: We will correct this.*

L70: You could add to the final sentence "and a potential driving force during the Pliocene".

*AC: We agree that this would be a good addition and we will add it to the final sentence.*

L94: Please provide some information about the spin-up times. I'm sure this is in earlier papers, but since this paper deals with the deep ocean, it seems appropriate to include it here.

*AC: We will add a column to Table 1 with the total runtime of all simulations, as also shown in Zhang et al. (2021).*

L234: Is this really a Gulf Stream variability issue? The single data point that is outside the model range is from Mg/Ca, and the Mg/Ca is also way out of the model range at the more stable site 609, where other proxies do fall within the model spread. So perhaps it is more of a Mg/Ca proxy issue?

*AC: It is indeed possible that this could be a Mg/Ca proxy issue. McClymont et al. (2020) raise several suggestions as to why there are large differences between their Mg/Ca and UK'37 SST reconstructions, including calibration differences and environmental factor such as seasonality. We will add "It may also be due to other factors such as calibration and seasonality affecting the Mg/Ca SST reconstructions (McClymont et al., 2020), since we see that the Mg/Ca proxy SST is much lower than the PlioMIP2 model range at location 609 as well." at line 237*

Section 3.1.2: About half of the models (6 of 13) do not show that the correlation between AMOC and SST is stronger at the mPWP, so the statement that the SST is more sensitive to the AMOC during the Pliocene could be misleading as a general statement. I suggest you add the mean (and perhaps STD) of the models' correlation coefficients so one could see if and where the correlations are stronger during the mPWP than the PI and also where they are positive and strong in particular. If nothing obviously pops up, then it illustrates clearly the model-dependancy of the correlations which is also important to keep in mind.

*AC: We agree that a figure showing the multi-model mean (MMM) and standard deviation of the models' correlation maps is useful. We will add this figure to the supplement and refer to it in our discussion. We find that in the MMM correlation map, the absolute correlation coefficient is smaller than in the individual maps due to spatial patterns varying across the models. However, the MMM correlation map does show a consistent pattern with relatively high correlation centered at 45N, 30W in both the PI and mPWP. In the mPWP, the extent of the area of positive correlation increases, and so do the mean and maximum correlation coefficients, centered around the same area. This supports our statement that the SST is more sensitive to the AMOC in the mid-Pliocene. We do think that it is important to stress the model-dependency of the correlation maps and will therefore modify lines 253-256.*

Figure 3 Caption: Shouldn't that be "95% confidence level"?

*AC: Yes, we will correct this.*

Section 3.2: While the AMOC and meridional density gradients are often correlated, the AMOC is not driven my meridional density gradients. Instead it is pressure, which combines density and the vertical stratification, that drives overturning (De Boer et al., 2010, doi.org/10.1175/2009JPO4200.1). The meridional pressure gradients can be estimated by delta_Rho*H^2 (meridional density gradient times the square of the depth of the max of overturning streamfunction). It may or may not make a huge difference here, but at least the text should be clear that a correlation with the density gradient is only expected if the depth of the overturning is not affected by processes that are not directly related to the strength of the AMOC (like AABW). Just like if the AMOC correlates

often with a meridional salinity gradient we would not make the general statement that the AMOC is driven by meridional salinity gradients.

*AC: Thank you for bringing this to our attention. We will change any statements about the AMOC being "driven" by density to being "affected" by density. We will also add your explanation given above concerning the influence of the depth of the overturning to section 3.2, referring to De Boer et al., 2010.*

Figure 4: Could you add a little space between the left and center column and center and right column to make it easier to figure out to which subplot the vertical text belongs? The caption says the density gradient is plotted against the AMOC anomaly. Is it not more conventional to say the y-axis is plotted against the axis? (As in dependent variable against independent variable?).

*AC: We will make the suggested changes to the figure and correct the caption.*

Figure 4 and L273->: As discussed in the general comment, it would be good to add the correlation coefficient here to compare with assumed forcing by freshwater changes. Also, could you add (in the supplement perhaps) the correlations of AMOC with the northern band of salinity/density and the correlations with the southern band salinity/density band separately to see whether the changes in the gradient originates more from the north or south? It could strengthen the argument that it all comes from the Bering Strait closure or it could force a rethink of that.

*AC: As mentioned in our reply to the general comment, we will do a correlation analysis on the freshwater changes and AMOC. We will also add a figure to the supplement that shows the suggested correlations with the northern and southern band separately. Our preliminary analysis shows that the density anomalies in both boxes correlate well with the AMOC but that it is only in the northern box that salinity anomalies show a significant correlation with the AMOC.*

L270 with regards to the GISS2.1G, on what evidence is the statement based that the deep water is formed north of 65N? When looking at the streamfunctions in Figure 1 in Zhang et al. (2021) this is not obvious, in fact, in other models the streamfunction stretches further north and the maximum overturning is at a similar latitude in GISS than the other models. From a quick inspection it seems that the GISS is the only model that has a maximum overturning deeper than 1km. Have you tried calculating the meridional gradients over a deeper layer?

*AC: When we inspect GISS2.1G's individual Atlantic zonal mean density anomaly, we find that there is a relatively dense anomaly compared to the rest of the Atlantic at latitudes north of 65N both at the surface and in deeper layers. In the 100-year mean mixed layer depth of GISS2.1G we can see that North Atlantic deep water formation extends to about 80N in the mid-Pliocene (this is not the case in the pre-industrial). We will add a figure showing GISS2.1G's zonal mean density and the Atlantic mixed layer depth to the supplement. Calculating the gradients over deeper layers has no significant effect on the meridional density gradient in GISS2.1G.*

L299: Perhaps change "into the Atlantic" here to "into the Arctic", since you are discussing changes in the Arctic salinity.

*AC: We will change this.*

Figure 7: Could you use the same y-axis scale for these FW transports so they are more easily comparable?

*AC: We will adjust the y-axis scale.*

Figure 8: Comparing Figure 4i and 12b suggest a relation between SSS and PmE in the North Atlantic, with models having a strong salinity gradient, like EC-Earth, also having large PmE anomalies and vice versa (like GISS and CESM1.2). The correlation is not great, but see my general point (2) that argue that this does not negate a role for PmE for salinity. On a visual point, the figure would be easier to read with another contour interval level for the salinity anomaly, and a split of the positive and negative contours into solid and dashed lines.

*AC: As mentioned in our reply to the general comment, we recognize that the role of PmE may be larger than realized at first and we will provide further analysis on the freshwater forcing in the North Atlantic. We will adjust the contour levels in Figure 8.*

L365: Is the higher PmE attributable to more evaputation from the higher? Presumably a lot if this rains out downwind?

*AC: As >80% of PlioMIP2 models show enhanced precipitation in large parts of the North Atlantic (Haywood et al., 2020), the higher PmE is indeed due to an increase in evaporation. We think that this increase in evaporation may be linked to the higher sea surface temperatures in the North Atlantic. We are not sure what the second question refers to.*

L413: What is the advantage of the pseudo-sverdrup transport analysis in Figure 11 over calculating the actual southward gyre transport, as in Figure S6? The lack of changes in the interior in Figure S6 quite is illuminating and arguably a more direct measure of the wind-driven transport.

*AC: The reason for including Figure 11 is to illustrate the lack of changes in the surface wind forcing of the subtropical gyre. There are studies that explicitly connect a decreased meridional temperature gradient to decreased gyre circulation due to changes in the atmospheric forcing of the gyres (Vallis and Farneti, 2009; Farneti and Vallis, 2013). We think that it is valuable to show that this suggested mechanism does not appear to play a role in the decreased OHT by the mid-Pliocene subtropical gyre in the PlioMIP2 models, even though we do see a decreased meridional temperature gradient at the surface of the Atlantic Ocean. Figure S6 supports that the virtually unchanged wind forcing at the surface of the subtropical gyre indeed corresponds to a lack of change in the gyre circulation itself. We will let this explanation reflect in our discussion.*

L440: In theory, if the gyre volume transports are the same, the heat and freshwater transport must by definition be 100% driven by east-west differences in T and S. The correlations should be stronger if you calculate it according the suggested method in the general comment 1, since you would then not include the superficial gyre contribution below the wind-driven layer into gyre heat and freshwater transport. And playing with the place where the zonal differences are taken might help. But in theory Figs 12a-d together with Figure 11 or Figure S6, should suffice to illustrate that increase gyre H/F transports are due to zonal T/S differences.

*AC: As mentioned in our reply to general comment 1, our analysis suggests that including the azonal term below the wind-driven layer in the gyre component does not significantly impact the magnitude of the gyre OHT component. This seems to be due to the meridional velocity field being quite zonally uniform below the wind-driven layer, and thus contributes significantly only to the overturning OHT component. As there are intermodel differences in the spatial structure of the temperature/salinity anomaly fields, it is impossible to find a perfect correlation when using the same longitudinal box for all models to define the gradient. However, the correlations we show are significant above*

*the 99% confidence level. We therefore think that our results are sufficient to illustrate that the increase in transports is due to zonal asymmetry in the temperature and salinity anomaly fields.*

L501-508: I suggest leaving out this description of the a-priori expectations. It's difficult to follow and turns out not to be substantiated by the results.

*AC: We think it is useful to explain that changes in OHT by the gyre do not result from changes in atmospheric forcing, even though the mid-Pliocene simulations show a warmer climate with reduced meridional temperature gradients. To enable readers to follow our line of thought more easily, we plan to rephrase L501-508 along the lines of:*

*"It appears that the opposing response of the subtropical gyre OHT component does not result from coupling between the atmospheric forcing of the gyre circulation and the reduced meridional temperature gradient over the mid-Pliocene Atlantic Ocean. Such a mechanism has been proposed by earlier studies such as Farneti and Vallis (2013). As there are no substantial changes in the gyre circulation itself, we find that changes in the ocean temperature and …" (continue from L509)*

L535: The closing of the Bering Strait is likely the driver of the AMOC changes, but the sentence "is thereby the driver" is arguably too strong, since it has not been shown in a rigorous analysis. For the pliocene runs, the freshwater input from the Arctic was certainly lower and could have driven the higher salinity, but also the PmE was also lower. The PmE did not correlate to the AMOC well, but that was an intermodel correlation. No similar correlation was attemped between the AMOC and the Bering Strait freshwater input for instance, or between PmE versus salinity and Arctic freshwater input versus salinity. Figures S3 for instance shows that in GISS and HadCM3, the Bering Strait freshwater input was really weak in the PI, yet they both have a stronger than average AMOC increase during the mPWP. So these things are still a bit complicated. Exploring this further may very well be beyond the scope of the current manuscript, in which case one could settle for a reasonable though somewhat more careful "plays a dicernable role". Similarly, lines 10-11 can be a bit more carefully stated.

*AC: Thank you for pointing this out. We will extend our freshwater input analysis and adjust our conclusions based on these results, as we have found that our earlier conclusion concerning the effect of the gateway closure on the AMOC may indeed be too strong.*

All in all this is a really interesting thought provoking paper.

*AC: We once more thank the reviewer very much for their constructive review.*

*References:*

*Hall, M. M., & Bryden, H. L. (1982). Direct estimates and mechanisms of ocean heat transport. Deep Sea Research Part A. Oceanographic Research Papers, 29(3), 339–359. https://doi.org/10.1016/0198-0149(82)90099-1.*

*Johns, W. E., Baringer, M. O., Beal, L. M., Cunningham, S. A., Kanzow, T., Bryden, H. L., Hirschi, J. J. M., Marotzke, J., Meinen, C. S., Shaw, B., & Curry, R. (2011). Continuous, Array-Based Estimates of Atlantic Ocean Heat Transport at 26.5°N. Journal of Climate, 24(10), 2429–2449. https://doi.org/10.1175/2010JCLI3997.1.*

*Jüling, A., Zhang, X., Castellana, D., von der Heydt, A. S., & Dijkstra, H. A. (2021). The Atlantic's freshwater budget under climate change in the Community Earth System Model*

with strongly eddying oceans. *Ocean Science, 17*(3), 729–754. https://doi.org/10.5194/os-17-729-2021.

Viebahn, J. P., von der Heydt, A. S., Le Bars, D., & Dijkstra, H. A. (2016). Effects of Drake Passage on a strongly eddying global ocean. *Paleoceanography, 31*(5), 564–581. https://doi.org/10.1002/2015PA002888.

Dowsett, H. J., Foley, K. M., Stoll, D. K., Chandler, M. A., Sohl, L. E., Bentsen, M., Otto-Bliesner, B. L., Bragg, F. J., Chan, W.-L., Contoux, C., Dolan, A. M., Haywood, A. M., Jonas, J. A., Jost, A., Kamae, Y., Lohmann, G., Lunt, D. J., Nisancioglu, K. H., Abe-Ouchi, A., … Zhang, Z. (2013). Sea Surface Temperature of the mid-Piacenzian Ocean: A Data-Model Comparison. *Scientific Reports, 3*(1), 2013. https://doi.org/10.1038/srep02013.

Haywood, A. M., Tindall, J. C., Dowsett, H. J., Dolan, A. M., Foley, K. M., Hunter, S. J., Hill, D. J., Chan, W.-L., Abe-Ouchi, A., Stepanek, C., Lohmann, G., Chandan, D., Peltier, W. R., Tan, N., Contoux, C., Ramstein, G., Li, X., Zhang, Z., Guo, C., … Lunt, D. J. (2020). The Pliocene Model Intercomparison Project Phase 2: Large-scale climate features and climate sensitivity. *Climate of the Past, 16*(6), 2095–2123. https://doi.org/10.5194/cp-16-2095-2020.

Zhang, Z., Li, X., Guo, C., Otterå, O. H., Nisancioglu, K. H., Tan, N., Contoux, C., Ramstein, G., Feng, R., Otto-Bliesner, B. L., Brady, E., Chandan, D., Peltier, W. R., Baatsen, M. L. J., von der Heydt, A. S., Weiffenbach, J. E., Stepanek, C., Lohmann, G., Zhang, Q., … Abe-Ouchi, A. (2021). Mid-Pliocene Atlantic Meridional Overturning Circulation simulated in PlioMIP2. *Climate of the Past, 17*(1), 529–543. https://doi.org/10.5194/cp-17-529-2021.

Vallis, G. K., & Farneti, R. (2009). Meridional energy transport in the coupled atmosphere-ocean system: Scaling and numerical experiments. *Quarterly Journal of the Royal Meteorological Society, 135*(644), 1643–1660. https://doi.org/10.1002/qj.498.

Farneti, R., & Vallis, G. K. (2013). Meridional Energy Transport in the Coupled Atmosphere–Ocean System: Compensation and Partitioning. *Journal of Climate, 26*(18), 7151–7166. https://doi.org/10.1175/JCLI-D-12-00133.1.

---

## Author Comment (AC2)

*AC: The authors would like to thank Anonymous Referee #2 for the comments and feedback. We will answer the questions and comments below and indicate the changes we plan to make in the revised manuscript.*

This study investigates the cause of the stronger AMOC in the PlioMIP2 ensemble and how Atlantic OHT and North Atlantic SSTs are influenced by this strengthening. In alignment with the findings of Otto-Bliesner et al., the authors conclude that the closure of the Bering Strait and Canadian Archipelago is the primary cause of the strengthening of the AMOC in the PlioMIP2 ensemble as it leads to a decrease in the southward freshwater transport into the Labrador Sea promoting high salinities and hence deep water formation. The authors go on to investigate the relationship between changes in the strength of the AMOC and changes in ocean heat transport. They find that AMOC driven increases in the ocean heat transport are partially compensated by a reduction of the gyre component, which helps to explain the model spread in ocean heat transport responses. I found the paper clearly written and analysis presented to be very thorough.

Major comments

When comparing simulated and reconstructed Pliocene SSTs Ln 110 mentions that an observational pre-industrial SST dataset is used to calculated the Pliocene minus Preindustrial SST change for the six different sites. If core top values are available they should be used rather than the pre-industrial SST dataset as using the former provides more of an apples with apples comparison and can have an impact e.g. The Haywood et al. (2020) analysis shows no Pliocene warming off California when differencing the Foley and Dowsett Uk37 records from the preindustrial SST dataset, but in Tierney et al., 2019 which differences these records from core top shows significant warming.

*AC: While it is an interesting idea to use core top values as pre-industrial SSTs, the time that is represented by the core top differs from core to core depending on local sedimentation rates. Therefore, the time that the core top value represents will be different for each site while PlioMIP2 experiments are temporally consistent, using prescribed forcings equivalent to 1850 for the pre-industrial experiment. We think that in this case it is a more consistent comparison to use the pre-industrial SST dataset for data-model comparison. In addition, performing a core top analysis would be beyond the scope of this study and could introduce additional complications in interpreting the difference between proxy data and model-derived SST anomalies.*

In establishing the mechanism responsible for the higher North Atlantic salinities in Fig. 5 I am not 100% convinced by the analysis that led to the statement "we find no indication that the PmE freshwater flux is the driving force behind the mid-Pliocene AMOC strength increase". While the analysis provided in Section 3.2.2 is thorough and provides a case for the role of reduced freshwater transport associated with the closure of the Canadian Archipelago (0.04 Sv), I do not find the analysis in Section 3.2.3 convincing in ruling out the role of changes in the Pliocene-Preindustrial North Atlantic surface freshwater flux. Fig. 8 shows that the MMM P-E is more negative over most of the subtropical gyre and parts of the subpolar gyre in the Pliocene simulations. This leads to more saline surface waters within the subtropical gyre (Figs. 4 & 5) that are advected into the subpolar region. Evidence of this is also seen in Fig. 6c if one takes the divergence (derivative) of the red and black lines respectively. The more rapid decline in the wind-driven northward freshwater transport (Fig. 6c, blue dashed line) is presumably due to enhanced surface evaporation due to warmer sea surface temperatures (particularly given the results of the gyre transport analysis in Section 3.4). This implies that another major cause of the higher Pliocene salinities is enhanced subtropical evaporation which is not adequality captured when using the local region defined in Fig. 8a (due to the role of advection). Therefore in addition to the closure of the Canadian Archipelago mechanism, perhaps the warmer north Atlantic SSTs due to more positive

regional radiative feedbacks or forcing changes in PlioMIP2 relative to PlioMIP1 might also help explain the stronger MMM AMOC result? An ocean salinity budget such as those conducted in Emile-Geay et al., 2003 and Burls et al., 2017 provide a framework that would account for the relative roles of freshwater transport and surface freshwater flux changes. While conducting such an analysis might be beyond the scope of the current study, the authors need to either provide an analysis that rules out a significant role for surface freshwater flux changes (e.g. converting mm/day x area to a Sv change over the subtropical and subpolar regions respectively to compare with the 0.04Sv change in the freshwater transport at 65N) or discuss this caveat in their interpretation accordingly.

Emile-Geay, M. A. Cane, N. Naik, R. Seager, A. C. Clement, A. van Geen, (2003) Warren revisited: Atmospheric freshwater fluxes and "Why is no deep water formed in the North Pacific." J. Geophys. Res. Atmos. 108, 3178–3112

Burls, N J, A V Fedorov, D M Sigman, S L Jaccard, R Tiedemann, and G H Haug, (2017) "Active Pacific Meridional Overturning Circulation (PMOC) during the Warm Pliocene." Science Advances 3, 9, e1700156.

*AC: We agree that our current analysis does not rule out a significant effect of the PmE on the North Atlantic salinity and should be supported by a more thorough analysis of the effect of changes in freshwater input (both via transport and surface fluxes) on the North Atlantic salinity. It is unfortunately not possible to perform an ocean salinity budget due to lack of data concerning the surface freshwater flux. However, using available data we will perform additional correlation analyses and elaborate on the respective effects of the altered Arctic freshwater influx and of PmE in the mid-Pliocene on the North Atlantic salinity in our revised manuscript. Preliminary results have shown that the role of the negative PmE anomaly in the high mid-Pliocene North Atlantic salinity may be greater than assumed before. We will therefore adjust our discussion and expect to nuance our conclusion about the gateway closure being the sole cause of the high North Atlantic salinity and AMOC strengthening.*

**Minor comments and suggestions:**

Ln 30-35: When comparing the response of the AMOC in future warming scenarios with Pliocene warming one has to take into account the timescales of the mechanisms involved and if the transient or equilibrium response is being assessed e.g. the short-term stratification of the Ocean which leads to AMOC decline in 21$^{st}$ century in projections will weaken as heat defuses downwards and there can be recovery as equilibrium is reached. A sentence or two acknowledging this subtility is needed here.

*AC: We will add a comment about this in the introduction.*

Eqn 1: I suggest changing vT to \bar{vT} to indicate that this is the 100-year mean of the product of v and T as opposed to the product of the 100-year mean of v and T respectively as in equ 3. Similarly for Ln 145, change vT to \bar{vT}

*AC: We will replace these two instances with \bar{vT} to indeed make clear that this is the 100-year mean of the product.*

Ln 167-168: Are COSMOS and HadCM3 the only two model for which the total OHT is inferred from the SHF? Is the \bar{vT} save and provided for all the other models?

*AC: Yes, COSMOS and HadCM3 are the only two models for which the total OHT is inferred from the SHF. The \bar{VT} is not part of the model output and provided for all*

*of the other models, but they have all provided the total OHT that has been calculated either on- or offline based on this product.*

Figure 2 caption: I am confused about the calculation of Fig. 2b is "with respect to the average MMM North Atlantic SST (30N-70N)" supposed to be "with respect to the average MMM North Atlantic SST (30N-70N) anomaly"?

*AC: Yes, we will change this into "with respect to the average MMM North Atlantic SST (30N-70N) anomaly"*

Section 3.1.2 and Fig. 3: This is for zero lag-lead and the annual times scale. The correlations between AMOC and SST could be stronger, and perhaps more similar, if the lag/lead of maximum correction for subpolar SST is shown? Also if a ten-year running mean is applied to filter out the influence of processes influencing SST associated with atmospheric noise.

*AC: A lag-lead analysis did not show significant differences in correlation patterns or strength to the results shown in Figure 3. In addition, it leads to complications in terms of models having different optimal lag-lead times. For consistency purposes, we have chosen to show the lag zero correlation. We have also performed the correlation analysis using a 10-year running mean. While such filtering generally increases the correlation coefficient, it does not significantly change the correlation patterns or the individual model trends in the correlations as presented in Table S2. As smoothing of the data causes complications when interpreting the significance and there are no qualitative changes to our results when using the 10-year running mean, we think it is best to show the correlation plots of the annual data.*